

# Non-minimal coupling, negative null energy, and effective field theory

Jackson R. Fliss[1,2], Ben Freivogel[1,3],
Eleni-Alexandra Kontou[1,3,4] and Diego Pardo Santos[4*]

**1** ITFA, Universiteit van Amsterdam, Science Park 904, 1098 XH Amsterdam, the Netherlands
**2** Department of Applied Mathematics and Theoretical Physics,
University of Cambridge, Cambridge CB3 0WA, United Kingdom
**3** GRAPPA, Universiteit van Amsterdam, Science Park 904,
1098 XH Amsterdam, the Netherlands
**4** Department of Mathematics, King's College London,
Strand, London WC2R 2LS, United Kingdom

⋆ diego.pardo@kcl.ac.uk

## Abstract

The non-minimal coupling of scalar fields to gravity has been claimed to violate energy conditions, leading to exotic phenomena such as traversable wormholes, even in classical theories. In this work we adopt the view that the non-minimal coupling can be viewed as part of an effective field theory (EFT) in which the field value is controlled by the theory's cutoff. Under this assumption, the average null energy condition, whose violation is necessary to allow traversable wormholes, is obeyed both classically and in the context of quantum field theory. In addition, we establish a type of "smeared" null energy condition in the non-minimally coupled theory, showing that the null energy averaged over a region of spacetime obeys a state dependent bound, in that it depends on the allowed field range. We finally motivate our EFT assumption by considering when the gravity plus matter path integral remains semi-classically controlled.



# 1 Introduction

Non-minimal coupling to gravity is a famous example where energy conditions are violated. Terms in the Lagrangian of the form

$$\delta \mathcal{L} = \xi R \phi^2,$$ (1)

allow violations of the Null Energy Condition (NEC) even at the classical level. Such terms have been claimed to have dramatic physical consequences, such as the construction of traversable wormholes in classical theories [1, 2]. However under the usual Wilsonian paradigm, terms of non-minimal coupling can appear natural in low-energy effective actions of a UV theory of quantum gravity and are common in models of inflation, with either positive or negative coupling constant (see e.g. Ref. [3] and Ref. [4], among many others). How do we then reconcile this with the apparent power of non-minimal coupling for generating troublesome geometries?

In this paper, we take the philosophy that

- The physically relevant question is whether non-minimal coupling can lead to exotic spacetime geometries. Since the non-minimal coupling term also modifies the Einstein equations, we must analyze either suitably defined 'effective' energy conditions or work in the 'Einstein frame', where the gravitational equations of motion take the standard form.

- The non-minimal coupling should be treated as the first term in an effective field theory (EFT) expansion that includes all terms in a higher-derivative and polynomial expansion in $\phi$. This EFT is sensible when there exists a cutoff on both large momenta *as well as large field values.* Here we work in the simplest EFT setting where a single UV cutoff $M_{\text{cutoff}}$ controls all quantities,

$$k_{\max}^2 \sim \phi_{\max}^{4/(n-2)} \sim M_{\text{cutoff}}^2,$$ (2)

where $k$ is the wavenumber and $n$ is the spacetime dimension.

Using this philosophy we revisit the effects of non-minimal coupling at both the classical and quantum levels. To be concrete, in this paper we focus on the simplest form of non-minimal

coupling, (1), however given the EFT framework mentioned directly above, this is most relevant term in both a derivative and field value expansion. Thus it is important to understand the effects of non-minimal coupling already at this first term. We will briefly comment on more general curvature couplings, when appropriate, at various points in the paper.

At the classical level, the 'effective NEC' can be violated by non-minimal coupling of either sign, so in this sense, non-minimally coupled theories are quite unusual. However, the effective Averaged Null Energy Condition (ANEC), which must be violated for certain traversable wormholes, is satisfied.

Despite these apparently exotic features, classically, non-minimal coupling can be eliminated by a combined conformal transformation and field redefinition, as we review in Sec. 2, roughly following [2]. The results of this section are not new. This field redefinition mixes the gravitational field and the scalar field and it takes us to the Einstein frame, where the NEC is obeyed. However, if the scalar field was initially free (aside from the gravitational coupling), the field redefinition introduces a self-interaction. The upshot is that, at the classical level, a non-minimally coupled theory can be mapped via a field redefinition to a completely standard theory of interacting scalar fields minimally coupled to gravity, which obeys standard energy conditions.

One can still ask whether exotic spacetimes can occur in the original 'Jordan frame'. In general, both descriptions are equally valid, however with differing operators, or probes, described by their respective fundamental fields. The question of which metric is the physical one depends on the probe one is interested in measuring. Some probes will follow geodesics of the Einstein frame, while others will follow geodesics of the Jordan frame. A key point, however, is that our EFT assumption enforces that the two metrics differ by a small amount. In particular, they are related by

$$\tilde{g}_{\mu\nu} \approx e^{-\frac{2}{n-2}(8\pi G_N \xi)\phi^2} g_{\mu\nu}. \tag{3}$$

Within the EFT framework, when $M^2_{\text{cutoff}}$ is Planckian or sub-Planckian, the quantity in the exponent is small and the two metrics are close. More generally, given the above assumption, we claim that the EFT can be treated in both the Einstein and Jordan frames with a clear map of probes between the frames.

In sections 3 and 4, we move to the quantum regime. In these sections, we do the analysis in the Jordan frame, where the non-minimal coupling is present. We demonstrate explicit quantum states that violate NEC by an arbitrarily large amount. These are not novel, since every quantum field theory allows for arbitrarily negative energy density at a point [5].

However, the non-minimally coupled (NMC) theory also violates smeared energy conditions, such as the double smeared null energy condition (DSNEC), which are satisfied by free theories. Similar results have been obtained for the energy density [6] and the effective energy density [7]. These kinds of smeared energy conditions or quantum energy inequalities (QEIs) are called *state dependent* as the bound depends on the state of interest. This is unlike the respective QEIs in the case of the minimally coupled theories.[1]

Given the genericity of non-minimal couplings it is important to understand the circumstances upon which negative null energy density is truly unbounded. As an important result of this work, we will show that the violations of smeared energy conditions can be controlled by invoking our EFT philosophy. Specifically we demonstrate that the violation of smeared energy conditions is driven by large field values: we require states with $\langle \phi^2 \rangle$ very large. We show that imposing a cutoff on the field value leads to a finite bound on the smeared null energy. Considering the null energy $T_{--}$ smeared over two null directions of length $\delta^+$ and

---

[1]In the case of the quantum strong energy inequality the state dependence appears also in the minimally coupled case with a state independent bound obtained only in the strictly massless case.

$\delta^-$, we establish a bound which can be schematically written as

$$\left\langle T^{\text{smear}}_{--} \right\rangle_\psi \geq -\frac{\mathcal{N}_n[\gamma]}{(\delta^+)^{n/2-1}(\delta^-)^{n/2+1}} - \frac{\#|\xi|\phi^2_{\max}}{(\delta^-)^2} = -\frac{\tilde{\mathcal{N}}_n[\gamma,\xi,\tilde{\phi}^2_{\max}]}{(\delta^+)^{n/2-1}(\delta^-)^{n/2+1}}\,. \tag{4}$$

Here $n$ is the space-time dimension, $\psi$ is the class of states of bounded field values, and $\phi^2_{\max}$ is the maximum value $\left|\left\langle\phi^2\right\rangle\right|$ can obtain within that class. $\mathcal{N}_n$ and $\tilde{\mathcal{N}}_n$ are dimensionless parameters depending on the details of the smearing function, the dimensionless mass, $\gamma = \delta^+\delta^- m^2$, and the dimensionless field cutoff $\tilde{\phi}^2_{\max} := (\delta^+\delta^-)^{\frac{n-2}{2}}\phi^2_{\max}$ (measured in units of the invariant smearing length). The above bound is for a single scalar field; at the level of analysis multiple scalar fields simply add, leading to an overall factor of the number of fields in the bound.

In Sec. 5, we generalize our classical construction of going to the Einstein frame to the quantum level. While we do not rigorously treat the corrections arising from the redefinition of the path integral measure, we provide arguments that a free NMC field can be mapped, through successive field redefinitions, to a minimally coupled field with a tower of self-interactions. We demonstrate explicitly how the potential $V(\phi)$ differs between the two frames. While the stress tensors in either frame are distinct operators, within the domain of the EFT quantum corrections are controlled and there is a clear map between these operators.

Lastly, we justify our EFT philosophy by considering the gravity + matter path integral in the original Jordan frame and ask when it is well-approximated by semi-classical computations. Focussing on positive coupling, $\xi > 0$, and making use of constrained instanton techniques, we establish that the gravitational theory becomes strongly coupled in this regime of large field values. This result provides a justification for our EFT requirement for $\xi \geq 0$. We do not have a similar argument for $\xi \leq 0$, although we offer an alternative (albeit less rigorous) signal of semi-classical breakdown for this case.[2] It would be of interest, particularly for models of inflation requiring large field values, to know what values of the non-minimal coupling can arise from a consistent UV theory and if this coupling can be constrained.

**Conventions**

Unless otherwise specified, we work in $n$ spacetime dimensions, assume $\hbar = c = 1$ and use metric signature $(-,+,\ldots,+)$. The D'Alembertian operator with respect to the metric $g$ is defined as $\Box_g := -g^{\mu\nu}\nabla_\mu\nabla_\nu$. The Riemann curvature tensor of $g$ is

$$R(X,Y)Z = \nabla_X\nabla_Y Z - \nabla_Y\nabla_X Z - \nabla_{[X,Y]}Z\,, \tag{5}$$

and the Ricci tensor $R_{\mu\nu}$ is its $(1,3)$-contraction. The Einstein Equation is $G_{\mu\nu} = 8\pi G_N T_{\mu\nu}$. The convention used for the metric, the Riemann tensor and the Einstein Equation is the $(+,+,+)$ according to Misner, Thorne and Wheeler [9].

When considering null subspaces in Minkowski we will denote, w.l.o.g., null coordinates[3] $x^\pm = t \pm x^1$ and transverse coordinates, $\vec{y} = (x^2,\ldots,x^n)$:

$$ds^2 = -dt^2 + \sum_{i=1}^{n}(dx^i)^2 = -dx^+dx^- + \sum_{a=2}^{n}(dy^a)^2\,. \tag{6}$$

Null derivatives will be denoted as $\partial_\pm := \frac{1}{2}(\partial_t \pm \partial_1)$. In momentum space this implies the following notation $k_\pm := \frac{1}{2}(k_0 \pm k_1)$; the inner product with coordinates remains unchanged, $k_\mu x^\mu = k_0 t + k_i x^i = k_+ x^+ + k_- x^- + k_a y^a$.

---

[2]Ref. [8] presents an argument demanding $\xi \geq 0$ under additional assumptions.

[3]Note importantly a discrepancy in integration measures $dt\,dx^1 = \frac{1}{2}dx^+dx^-$. In the interest of comparison to previous results and to be clear on this front, we will always denote integrations with respect to null-coordinates by $d^2x^\pm := dx^+dx^-$. Similarly integrations in momentum space will follow a similar notation: $d^2k_\pm := dk_+dk_- = \frac{1}{2}dk_0dk_1$.

For the Fourier transform we use the following convention

$$\tilde{f}(k) = \int_{\mathbb{R}^n} d^n x f(x) e^{ikx} = \int dt\, dx^1\, d^{n-2}\vec{y}\, f(x) e^{ikx}\,. \tag{7}$$

## 2 Classical energy conditions

In this section we review the status of classical energy conditions in non-minimally coupled theories. We further present the transformation that brings the non-minimally coupled action (Jordan frame) to the minimally coupled one (Einstein frame).

### 2.1 The non-minimally coupled scalar field

The classical gravitational action integral for a non-minimally coupled scalar is

$$S = \int d^n x \sqrt{-g} \left[ \frac{(R-2\Lambda)}{16\pi G_N} - \frac{1}{2}(\nabla\phi)^2 - \frac{1}{2}\xi R\phi^2 - V(\phi) \right], \tag{8}$$

where $(\nabla\phi)^2 = g^{\mu\nu}(\nabla_\mu\phi)(\nabla_\nu\phi)$, $V(\phi)$ is the potential, and $\xi$ a dimensionless coupling constant.[4] The conformal coupling in $n$-dimensions is

$$\xi_c = \frac{n-2}{4(n-1)}, \tag{9}$$

so for four-dimensions $\xi_c = 1/6$.

For the massive, free scalar which will be studied in this work, we have $V(\phi) = m^2\phi^2/2$. The stress-energy tensor is obtained by varying the action (8):

$$T_{\mu\nu} = (\nabla_\mu\phi)(\nabla_\nu\phi) - \frac{1}{2}g_{\mu\nu}(m^2\phi^2 + (\nabla\phi)^2) + \xi(-g_{\mu\nu}\Box_g - \nabla_\mu\nabla_\nu + G_{\mu\nu})\phi^2. \tag{10}$$

What is interesting to note here is that the stress-energy tensor differs from minimal coupling even for vanishing curvature, $G_{\mu\nu} = 0$. In this case the additional terms proportional to $\xi$ can be viewed as "improvement terms" to the canonical stress-tensor. 'On shell', $\phi$ obeys the field equation

$$\left(\Box_g + m^2 + \xi R\right)\phi = 0\,. \tag{11}$$

Using the identity

$$\phi\Box_g\phi = \frac{1}{2}\Box_g\phi^2 - (\nabla\phi)^2\,, \tag{12}$$

we can write

$$\begin{aligned} T_{\mu\nu} = (1-2\xi)(\nabla_\mu\phi)(\nabla_\nu\phi) - \frac{1}{2}(1-4\xi)g_{\mu\nu}(m^2\phi^2 + \xi R\phi^2 + (\nabla\phi)^2) \\ - 2\xi\left(\phi\nabla_\mu\nabla_\nu\phi + \frac{1}{2}R_{\mu\nu}\phi^2\right). \end{aligned} \tag{13}$$

We are interested in the null energy which classically is

$$\rho_n \equiv T_{\mu\nu}\ell^\mu\ell^\nu = (1-2\xi)(\ell^\mu\nabla_\mu\phi)(\ell^\nu\nabla_\nu\phi) - 2\xi\left(\phi(\ell^\mu\ell^\nu\nabla_\mu\nabla_\nu\phi) + \frac{1}{2}R_{\mu\nu}\ell^\mu\ell^\nu\phi^2\right), \tag{14}$$

---

[4]The $\frac{1}{2}\xi R\phi^2$ coupling can be thought of as the first term in a field expansion of a generic class of curvature couplings of the form $\mathcal{A}[\phi]R$.

where $\ell^\mu$ is a null vector.

Alternatively, we can define an *effective stress tensor* by separating the curvature terms from the field terms in the Einstein equation. So using the Einstein Equation and Eq. (10) we have

$$G_{\mu\nu} = 8\pi G_N T^{\text{eff}}_{\mu\nu}, \tag{15}$$

where

$$
T^{\text{eff}}_{\mu\nu} = \frac{1}{1-8\pi G_N \xi \phi^2}\bigg( (\nabla_\mu \phi)(\nabla_\nu \phi) - \frac{1}{2}g_{\mu\nu}\Big(m^2\phi^2 + (\nabla\phi)^2 + \frac{\Lambda}{4\pi G_N}\Big) \\
+ \xi(-g_{\mu\nu}\Box_g - \nabla_\mu\nabla_\nu)\phi^2 \bigg). \tag{16}
$$

The null energy density for the effective stress tensor is

$$\rho^{\text{eff}}_n = \frac{1}{1-8\pi G_N \xi \phi^2}\left( (\ell^\mu\nabla_\mu\phi)(\ell^\nu\nabla_\nu\phi) - \xi(\ell^\mu\ell^\nu\nabla_\mu\nabla_\nu)\phi^2 \right). \tag{17}$$

For the purposes of constraining classical spacetimes allowed by the Einstein equation, it is useful to state energy conditions in terms of the effective stress tensor, as it is the quantity directly connected to the geometry.

It is evident from the form of Eqs. (15) and (16) that, for $\xi > 0$, the field $\phi$ experiences a critical value at $(8\pi G_N \xi)^{-1/2}$ by which stress-tensor changes sign. A change of sign of the coefficient of the stress-tensor means the change of the sign of the Einstein equation. Depending on the definition, this can also be considered as the change of the sign of the effective Newton constant. This is an important observation as we will see significant violations of effective average null energy conditions occur only when the field value is unbounded.

## 2.2 NEC and ANEC

In this section we examine the null energy conditions for the classical non-minimally coupled scalar and in particular the NEC and ANEC partly following [2].

The null energy for the non-minimally coupled scalar admits negative values even for flat spacetimes as is evident from Eqs. (14) and (17). But the situation is different for ANEC. Integrating Eq. (14) on an entire null geodesic $\gamma$ parametrized by $\lambda$ for vanishing curvature gives

$$\int \rho_n d\lambda = \int (\ell^\mu\nabla_\mu\phi)(\ell^\nu\nabla_\nu\phi)d\lambda - \xi\int \ell^\mu\ell^\nu\nabla_\mu\nabla_\nu(\phi^2)d\lambda. \tag{18}$$

The first term is non-negative and the second a total derivative. Assuming that the field has asymptotically vanishing derivatives, the ANEC is obeyed.

Turning to the effective stress-energy tensor we notice that the effective NEC can also be violated as evident by Eq. (17). On $\gamma$ we have

$$\rho^{\text{eff}}_n = \frac{1}{1-8\pi G_N \xi \phi^2}\left( \Big(\frac{d\phi}{d\lambda}\Big)^2 - \xi\frac{d^2(\phi^2)}{d\lambda^2} \right). \tag{19}$$

From this form we can find the cases that violate the NEC. For $\xi < 0$ any local maximum of $\phi^2$ violates the NEC. For $\xi > 0$ and small field values $8\pi\xi G_N\phi^2 < 1$ any local minimum of $\phi^2$ is a violation while for large field values $8\pi\xi G_N\phi^2 > 1$ any local maximum is a violation.

The "effective ANEC" integral for the effective stress-tensor is

$$\int_\gamma \rho^{\text{eff}}_n d\lambda = \int_\gamma d\lambda \frac{1}{1-8\pi G_N \xi \phi^2}\left( \Big(\frac{d\phi}{d\lambda}\Big)^2 - \xi\frac{d^2(\phi^2)}{d\lambda^2} \right). \tag{20}$$

Integrating by parts the last term

$$\int_\gamma \rho_n^{\text{eff}} d\lambda = \int_\gamma d\lambda \frac{1 - 8\pi\xi G_N(1-4\xi)\phi^2}{(1-8\pi G_N\xi\phi^2)^2}\left(\frac{d\phi}{d\lambda}\right)^2, \tag{21}$$

where we discarded the boundary terms, assuming a smooth geodesic. Then it is obvious that the effective ANEC integrand is non-negative for $\xi < 0$ and $\xi > 1/4$. For $0 < \xi < 1/4$ it is positive for small field values, $8\pi\xi(1-4\xi)G_N\phi^2 < 1$, and negative for large values. Of particular interest is the case where $8\pi G_N\xi\phi^2$ is close to 1 and the integrand negative because then we can have large effective ANEC violations. As mentioned above, this can lead to traversal and causality violating wormholes [1, 2, 10]. Interestingly, one can check that the positivity of the effective ANEC integrand is equivalent to a modified "Jordan frame NEC" proposed in [11].

## 2.3 Einstein and Jordan frames

The gravitational action integral for a non-minimally coupled scalar Eq. (8) can be brought to minimally coupled form by a conformal transformation along with a field redefinition

$$\tilde{g}_{\mu\nu} = \Omega^2 g_{\mu\nu}, \qquad \tilde{\phi} = F(\phi), \tag{22}$$

where $F$ is a real function. The conformal factor, $\Omega$, has a functional dependence on the scalar field $\Omega(\phi(x))$. This transformation brings Eq. (8) to an action with canonical kinetic terms for the metric and the scalar field, plus a new potential:

$$S = \int d^n x \sqrt{-\tilde{g}}\left[\frac{\tilde{R}}{16\pi G_N} - \frac{1}{2}(\tilde{\nabla}\tilde{\phi})^2 - \tilde{V}(\tilde{\phi})\right], \qquad \tilde{V}(\tilde{\phi}) = \Omega^{-n}(\phi)\left(\frac{\Lambda}{8\pi G_N} + V(\phi)\right), \tag{23}$$

where we regard $\phi = F^{-1}(\tilde{\phi})$ above. While above we have focussed on the simplest form of non-minimal coupling, $\frac{1}{2}\xi R\phi^2$, in Appendix A, we show this that such a transformation applies to a more general class of couplings of the form $\mathcal{A}[\phi]R$. As a simple example, for massless free scalars in asymptotically flat spacetimes, the conformal transformation gives an action with $\tilde{V}(\tilde{\phi}) = 0$. We will revisit the details of this field redefinition in the context of the quantum theory in section 5.

The frame where we have non-minimal coupling is called *Jordan frame* while the frame where we have minimal coupling *Einstein frame*. The conformally transformed action (23) leads to a stress-energy tensor

$$\tilde{T}_{\mu\nu} = (\tilde{\nabla}_\mu\tilde{\phi})(\tilde{\nabla}_\nu\tilde{\phi}) - \frac{1}{2}\tilde{g}_{\mu\nu}(2\tilde{V}(\tilde{\phi}) + (\tilde{\nabla}\tilde{\phi})^2). \tag{24}$$

Importantly all information about the conformation transformation appears in the $\tilde{V}(\tilde{\phi})$ of the stress-tensor. Then the null energy is

$$\tilde{\rho}_n = (\ell^\mu\tilde{\nabla}_\mu\tilde{\phi})(\ell^\nu\tilde{\nabla}_\nu\tilde{\phi}), \tag{25}$$

thus always obeying the NEC classically. The two frames are evidently not equivalent in terms of classical energy conditions as the NEC in the Einstein frame is obeyed while in the Jordan frame there are violations.

Extending this argument from classical to quantum field theory is not straightforward. One primary result of this paper is establishing when the above manipulations remain sensible in the quantum theory: This analysis will be performed in section 5.

Some authors (e.g. [12]) have argued that the fact that the field obeys the NEC in the Einstein frame means that this is the physical one. In particular, the stability of the classical

system in the Jordan frame is questioned due to the violation of energy conditions. The equivalence of the two frames is also questioned with a detailed discussion and literature search presented in [13].

As discussed in the introduction, we take the position that the physically relevant question is whether non-minimal coupling can lead to exotic spacetime geometries. For example, are wormholes allowed classically in the Jordan frame while not allowed in the Einstein one? Short wormholes, those allowing for causality violations, can only be constructed and sustained when the achronal ANEC is violated [10]. As we showed, the self-consistent, effective, ANEC is only violated when we have large field values, an unphysical effect if we view non-minimal coupling as an EFT.

Long wormholes such as the one in Ref. [14] do not require the violation of the achronal ANEC as there are no achronal null geodesics passing through their throat. While there is no relevant theorem, it is obvious that the violation of effective ANEC is needed to sustain long wormholes. Considering only cases where the connected areas are asymptotically flat, the divergence of null geodesics passing through the throat requires the violation of average null convergence condition and thus the effective ANEC. This point requires further consideration, but so far it seems that it is impossible to construct traversable wormholes in the Jordan frame without unphysical field values. In that sense the two frames can be considered equivalent.

# 3 An example of large negative null energy density in quantum field theory

Before we prove the form of a general energy inequality, it is useful to see how a non-minimal coupling can generate potentially troublesome energy densities in quantum field theory. In fact, this is simple to illustrate even for free theories in Minkowski space: The flat-space limit of (10) retains a dependence on the coupling $\xi$. This contribution to $T_{\mu\nu}$ is natural in the free theory where we can view it as an improvement term to the canonical stress tensor. As a corollary, what follows in the next two sections can also be regarded as a illustration of the failure of energy conditions under the addition of improvement terms to the stress tensor.

For this example, we will focus on the null stress tensor, $T_{--} = \ell_-^\mu \ell_-^\nu T_{\mu\nu}$, in a massless theory in flat-space. Much in this section follows the example given in Ref. [6], however generalized to the null stress tensor and with an extended discussion. In a Minkowski background we are free to renormalise the flat-space limit of (10) by normal ordering with respect to Fock modes:[5]

$$T_{--}(x) =: \partial_-\phi\partial_-\phi : (x) - \xi \left(: \partial_-^2\phi\phi : (x) + 2 : \partial_-\phi\partial_-\phi : (x) + : \phi\partial_-^2\phi : (x)\right). \tag{26}$$

We will null quantize $\phi$ as

$$\phi(x^+, x^-, \vec{y}_\perp) = \int_0^\infty \frac{dk_-}{2\pi\sqrt{k_-}} \int \frac{d^{n-2}p_\perp}{(2\pi)^{n-2}} \left( a_{k_-,\vec{p}_\perp} e^{ik_-x^- + i\frac{p_\perp^2}{4k_-}x^- + i\vec{p}_\perp\cdot\vec{y}_\perp} + \text{h.c.} \right), \tag{27}$$

with commutators

$$[a_{k_-,\vec{p}_\perp}, a^\dagger_{k'_-,\vec{p}'_\perp}] = (2\pi)^{n-1}\delta(k_- - k'_-)\delta^{n-2}(\vec{p}_\perp - \vec{p}'_\perp). \tag{28}$$

The vacuum, $|\Omega\rangle$, is annihilated by $a_{k_-,\vec{p}_\perp}$ and so normal-ordering places all $a$'s to the right. It will be useful to write $T_{--} = T_{--}^{(0)} + \delta T_{--}$ where $T_{--}^{(0)} = T_{--}|_{\xi=0}$ is the canonical stress-tensor and $\delta T_{--}$ is the improvement term proportional to $\xi$.

---

[5]This is equivalent to Hadamard renormalizing $T_{--}$ using the Minkowski vacuum as the reference state. We will discuss general renormalization schemes in section 4.1.

### 3.1 One-particle states with negative null energy

We now consider the following 1-particle state

$$|h_{\alpha_-,\alpha_\perp}\rangle := a^\dagger_{h_{\alpha_-,\alpha_\perp}}|\Omega\rangle\,, \qquad a^\dagger_{h_{\alpha_-,\alpha_\perp}} = \int_0^\infty \frac{dk_-}{2\pi\sqrt{k_-}}\int\frac{d^{n-2}p_\perp}{(2\pi)^{n-2}}h_{\alpha_-,\alpha_\perp}(k_-,\vec{p}_\perp)a^\dagger_{k_-,\vec{p}_\perp}\,, \quad (29)$$

where

$$h_{\alpha_-,\alpha_\perp}(k_-,\vec{p}_\perp)=\gamma\frac{\sqrt{k_-}\left(\frac{3}{2}\alpha_--k_-\right)}{\alpha_-^{3/2}\alpha_\perp^{\frac{n-2}{2}}}e^{-k_-/\alpha_--|\vec{p}_\perp|/\alpha_\perp}\,, \quad (30)$$

for positive parameters $\alpha_-$ and $\alpha_\perp$, and the normalization

$$\gamma=\frac{2^{\frac{n+1}{2}}(2\pi)^{\frac{n-1}{2}}}{\sqrt{5V_{n-3}\Gamma(n-2)}}\,, \qquad V_{n-3}=\text{volume of unit }S^{n-3}=\frac{2\pi^{\frac{n-2}{2}}}{\Gamma(\frac{n-2}{2})}\,, \quad (31)$$

chosen such that

$$\left[a_{h_{\alpha_-,\alpha_\perp}},a^\dagger_{h_{\alpha_-,\alpha_\perp}}\right]=1\,. \quad (32)$$

These states are similar to the ones considered in Ref. [6] for the energy density. From here on we will simply notate $h_{\alpha_-,\alpha_\perp}\equiv h_\alpha$ (and similarly for the oscillators and the state) unless there is a need to specifically demarcate $\alpha_-$ from $\alpha_\perp$.

The expectation value of $T_{--}$ at the spacetime origin, $x=0$, is simple to find. The coefficients in (30) have been chosen so that $\langle T^{(0)}_{--}(x=0)\rangle_{h_\alpha}$ vanishes and only the $\xi$ terms contribute:

$$\langle T_{--}(0)\rangle_{h_\alpha}=2\xi\int_0^\infty\frac{dk_-dk'_-}{(2\pi)^2k_-k'_-}\int\frac{d^{n-2}p_\perp d^{n-2}p'_\perp}{(2\pi)^{2(n-2)}}(k_--k'_-)^2h_\alpha(k_-,\vec{p}_\perp)h_\alpha(k'_-,\vec{p}'_\perp)$$

$$=-\frac{12V_{n-3}\Gamma(n-2)}{5\pi^{n-2}}\xi\,\alpha_-^2\alpha_\perp^{n-2}\,. \quad (33)$$

Let us focus foremostly on the case where $\xi>0$ (we will make remarks about the opposite case below). Then the expectation value is negative and with a magnitude controlled by the arbitrary parameters $\alpha_-$ and $\alpha_\perp$. This is consistent with lack of pointwise lower bounds on null energy in QFT [5]. We should expect instead that averaged over an appropriate neighborhood, $T_{--}$ is better behaved.

Continuity of the expectation value implies that there exists a finite neighborhood $\mathcal{U}$ containing $x=0$ such that

$$\langle T_{--}(x)\rangle_{h_\alpha}\leq-\frac{6V_{n-3}\Gamma(n-2)}{5\pi^{n-2}}\xi\,\alpha_-^2\alpha_\perp^{n-2}\,,\quad\forall\ x\in\mathcal{U}\,. \quad (34)$$

Furthermore noting

$$\left|\langle T_{--}(x)\rangle_{h_\alpha}\right|\leq\left|\langle T^{(0)}_{--}(x)\rangle_{h_\alpha}\right|+\left|\langle\delta T_{--}(x)\rangle_{h_\alpha}\right|\,, \quad (35)$$

where the above terms can be separately bounded above by pulling the absolute value into the momentum integrals, we can put the null energy density in the window

$$-\frac{6V_{n-3}\Gamma(n-2)}{5\pi^{n-2}}\xi\,\alpha_-^2\alpha_\perp^{n-2}\geq\langle T_{--}(x)\rangle_{h_\alpha}\geq-\frac{8V_{n-3}\Gamma(n-2)}{5\pi^{n-1}}\left(c_0+c_\xi\,\xi\right)\alpha_-^2\alpha_\perp^{n-2}\,,\quad\forall\ x\in\mathcal{U}\,, \quad (36)$$

where $c_0$ and $c_\xi$ are $O(1)$ numerical constants[6] independent of dimension and $\alpha_{-,\perp}$:

$$c_0 \approx 0.672\,, \qquad c_\xi \approx 5.786\,. \tag{38}$$

The above result implies that while $\langle T_{--}(x)\rangle_{h_\alpha}$ is bounded below (this lower bound above in fact applies for all spacetime points), it must be sufficiently negative for some neighborhood $\mathcal{U}$ about $x = 0$. Let us briefly comment on the spacetime extent of this neighborhood and its relation to the state parameters $(\alpha_-, \alpha_\perp)$. Our rough intuition is the size of neighborhoods satisfying (34) should decrease as $\alpha_-$ or $\alpha_\perp$ increase. To see that this is indeed true we note the following scaling properties of $h_{\alpha_-, \alpha_\perp}$:

$$
\begin{aligned}
h_{\lambda\alpha_-, \alpha_\perp}(k_-, \vec{p}_\perp) &= h_{\alpha_-, \alpha_\perp}(k_-/\lambda, \vec{p}_\perp) & \text{(Boost),} \\
h_{\lambda\alpha_-, \lambda\alpha_\perp}(k_-, \vec{p}_\perp) &= \lambda^{-\frac{d-2}{2}} h_{\alpha_-, \alpha_\perp}(k_-/\lambda, \vec{p}_\perp/\lambda) & \text{(Dilatation),}
\end{aligned} \tag{39}
$$

which imply, respectively,

$$
\begin{aligned}
\langle h_{\lambda\alpha_-, \alpha_\perp}|T_{--}(x^-, x^+, \vec{y}_\perp)|h_{\lambda\alpha_-, \alpha_\perp}\rangle &= \lambda^2 \langle h_{\alpha_-, \alpha_\perp}|T_{--}(\lambda x^-, \lambda^{-1}x^+, \vec{y}_\perp)|h_{\alpha_-, \alpha_\perp}\rangle\,, \\
\langle h_{\lambda\alpha_-, \lambda\alpha_\perp}|T_{--}(x^-, x^+, \vec{y}_\perp)|h_{\lambda\alpha_-, \lambda\alpha_\perp}\rangle &= \lambda^d \langle h_{\alpha_-, \alpha_\perp}|T_{--}(\lambda x^-, \lambda x^+, \lambda\vec{y}_\perp)|h_{\alpha_-, \alpha_\perp}\rangle\,.
\end{aligned} \tag{40}
$$

In particular the second relation implies that

$$
\begin{aligned}
-\lambda^d \frac{6V_{n-3}\Gamma(n-2)}{5\pi^{n-2}} \xi\, \alpha_-^2 \alpha_\perp^{n-2} \geq &\langle T_{--}(x)\rangle_{h_{\lambda\alpha}} \\
&\geq -\lambda^d \frac{8V_{n-3}\Gamma(n-2)}{5\pi^{n-1}} \left(c_0 + c_\xi\, \xi\right) \alpha_-^2 \alpha_\perp^{n-2}\,, \quad \forall\ x \in \lambda^{-1}\mathcal{U}\,.
\end{aligned} \tag{41}
$$

That is, one can push the window in which $\langle T_{--}\rangle_{h_\alpha}$ lies to lower negative values by scaling $(\alpha_-, \alpha_\perp)$ at the cost of inversely scaling the size of the neighborhood on which (36) holds. This is illustrated in figure 1. However we now show that we can push the null-energy density arbitrarily low by considering multi-particle states.

## 3.2 Multi-particle states

Now we consider the state

$$|(h_\alpha)^N\rangle := \frac{1}{\sqrt{N!}} \left(a_{h_\alpha}^\dagger\right)^N |\Omega\rangle\,. \tag{42}$$

The null energy, being quadratic in creation and annihilation operators, then scales with $N$:

$$\langle (h_\alpha)^N | T_{--}(x) | (h_\alpha)^N \rangle = N \langle h_\alpha | T_{--}(x) | h_\alpha \rangle\,. \tag{43}$$

This is true for all $x$ and so it follows that given the same parameters $(\alpha_-, \alpha_\perp)$ the same neighborhood, $\mathcal{U}$, and same constants, $c_{0,\xi}$, as in (36)

$$
\begin{aligned}
-N \frac{6V_{n-3}\Gamma(n-2)}{5\pi^{n-2}} \xi\, \alpha_-^2 \alpha_\perp^{n-2} \geq &\langle T_{--}(x)\rangle_{(h_\alpha)^N} \\
&\geq -N \frac{8V_{n-3}\Gamma(n-2)}{5\pi^{n-1}} \left(c_0 + c_\xi\, \xi\right) \alpha_-^2 \alpha_\perp^{n-2}\,, \quad \forall\ x \in \mathcal{U}\,.
\end{aligned} \tag{44}
$$

---

[6]More precisely:

$$c_0 = \left(\int_0^\infty d\kappa \sqrt{\kappa}\, |3/2 - \kappa|e^{-\kappa}\right)^2\,, \qquad c_\xi = \int_0^\infty d\kappa_1 d\kappa_2 \frac{(\kappa_1 - \kappa_2)^2}{\sqrt{\kappa_1\kappa_2}} |3/2 - \kappa_1||3/2 - \kappa_2|e^{-\kappa_1 - \kappa_2}\,. \tag{37}$$

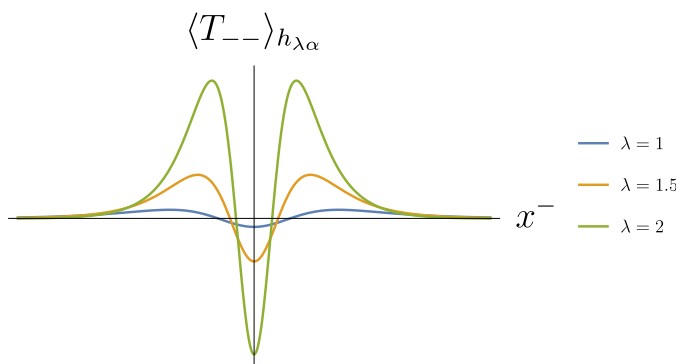

Figure 1: $\langle T_{--}\rangle_{h_{\lambda\alpha}^{(1)}}$ in $n = 4$ dimensions at the point $(x^-, x^+ = \vec{y}_\perp = 0)$ plotted (in units of $\alpha_-^2 \alpha_\perp^2$) along $x^-$. The coupling has been set to the conformal value $\xi = \frac{1}{6}$. As the parameters of the state are scaled up, the minimum at $x^- = 0$ is made more negative; correspondingly, the window in $x^-$ in which $\langle T_{--}\rangle_{h_{\lambda\alpha}}$ is negative also shrinks.

Now suppose that we fix a spacetime neighborhood, $\bar{\mathcal{U}}$, around $x = 0$. Then there exist a series of $N$-particle states with parameters $(\bar\alpha_-, \bar\alpha_\perp)$ such that

$$-N\frac{6V_{n-3}\Gamma(n-2)}{5\pi^{n-2}}\,\xi\,\bar\alpha_-^2\,\bar\alpha_\perp^{n-2} \geq \langle T_{--}(x)\rangle_{(h_{\bar\alpha})^N}$$
$$\geq -N\frac{8V_{n-3}\Gamma(n-2)}{5\pi^{n-1}}\left(c_0 + c_\xi\,\xi\right)\bar\alpha_-^2\,\bar\alpha_\perp^{n-2}, \quad \forall\ x\in\bar{\mathcal{U}}, \tag{45}$$

for any $N$. For any $\bar\rho > 0$ we can choose the particle number

$$\bar{N} = \left\lceil \frac{5\pi^{n-2}}{6\xi\,V_{n-3}\Gamma(n-2)}\frac{\bar\rho}{\bar\alpha_-^2\,\bar\alpha_\perp^{n-2}} \right\rceil \tag{46}$$

(where $\lceil\ldots\rceil$ is the ceiling function), then we will have

$$-\bar\rho \geq \langle T_{--}(x)\rangle_{(h_{\bar\alpha})^N} \geq -(c_0 + c_\xi\xi)\left(\frac{4}{3\pi}\frac{\bar\rho}{\xi} + \frac{8V_{n-3}\Gamma(n-2)}{5\pi^{n-1}}\alpha_-^2\alpha_\perp^{n-2}\right), \quad \forall\ x\in\bar{\mathcal{U}}. \tag{47}$$

What we have found is that given a spacetime region of arbitrary size, and an arbitrary $\bar\rho$, we can find a finite-particle-number state such that $\langle T_{--}\rangle$ is upper-bounded by $-\bar\rho$. Thus, in contrast to the canonical stress tensor, Eq. (47) implies the non-existence of a lower bound on the smeared null energy for the improved $T_{--}$. As we will show later, this conclusion is not special to the improved stress tensor in Minkowski space: it generically extends to non-minimally coupled scalar fields. However the need for large particle number has implications for other observables besides the energy density.

Lastly let us briefly point out that though we have assumed $\xi > 0$ above, the case where $\xi < 0$ can also exhibit pathologies in these same class of multi-particle states. In particular, plotting $\langle T_{--}\rangle_{h_{\lambda\alpha}}$ in this case, we see that for sufficiently large values of $|\xi|$ the null-energy density at $x^+ = \vec{y}_\perp = 0$ can dip into negative regions. These regions can be enhanced, via the same arguments above, to being arbitrarily negative by increasing the particle number.

For small values of $|\xi|$ it is still possible to find regions of negative null energy density by generalizing Eq. (30) by a complex parameter $\zeta$:

$$h_{\alpha_-,\alpha_\perp}^{(\zeta)}(k_-,\vec{p}_\perp) = \gamma_\zeta\frac{\sqrt{k_-}(\zeta\,\alpha_- - k_-)}{\alpha_-^{3/2}\alpha_\perp^{\frac{n-2}{2}}}e^{-k_-/\alpha_- - |\vec{p}_\perp|/\alpha_\perp}, \tag{48}$$

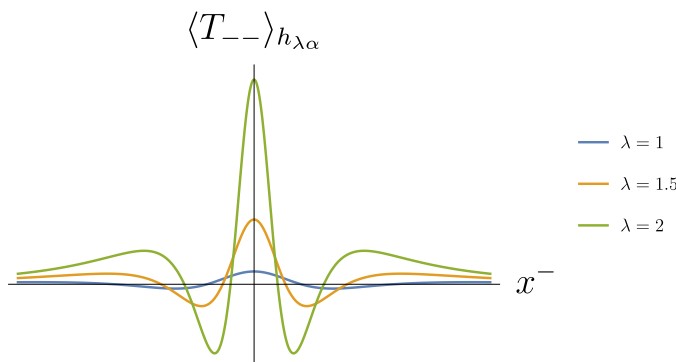

Figure 2: $\langle T_{--}\rangle_{h_{\lambda\alpha}}$ in $n = 4$ dimensions at the point $(x^-, x^+ = \vec{y}_\perp = 0)$ plotted (in units of $\alpha_-^2 \alpha_\perp^2$) along $x^-$. The coupling has been set to $\xi = -\frac{1}{4}$. Though $\langle T_{--}\rangle_{h_{\lambda,\alpha}}$ is positive at $x^- = 0$, there exist regions of negative null energy which can be amplified by large particle number.

and tuning $\zeta$ appropriately. Here $\gamma_\zeta$ chosen such that $a^\dagger_{h^{(\zeta)}_{\alpha_-,\alpha_\perp}}$ and $a_{h^{(\zeta)}_{\alpha_-,\alpha_\perp}}$ (defined analogously to Eq. (29)) obey normalized commutation relations à la Eq. (32). This is illustrated in figure 3. Again, once a region of negative null energy density exists it can be enhanced to arbitrary magnitude by increasing the particle number. The existence of states of arbitrarily negative null energy density, regardless of the sign of coupling, is consistent with the results of Section 4.

## 3.3 Large negative null energies require large field values

The pointwise expectation value of $: \phi^2 :$ in these class of states $|h^{(\zeta)}_{\alpha_-,\alpha_\perp}\rangle$ (again, defined by functions Eq. (48)) also scales with $N$:

$$\langle : \phi^2(x) : \rangle_{\left(h^{(\zeta)}_\alpha\right)^N} = N\langle : \phi^2(x) : \rangle_{h^{(\zeta)}_\alpha}. \tag{49}$$

Engineering states with large negative null energy by increasing the particle number has the effect of amplifying fluctuations of the field values.

Let us suppose there existed a bound on field values in an effective field theory:

$$\left|\langle : \phi^2(x) : \rangle_\psi\right| \leq \phi^2_{\max}, \tag{50}$$

for any state, $\psi$, and any point in spacetime. We will try to motivate such a bound in section 5. This implies a maximum particle number for $h$-states and thus a lower bound on negative null-energy. To illustrate this, let us focus on $\xi > 0$ and use $h^{(\zeta)}_{\alpha_-,\alpha_\perp}\Big|_{\zeta=3/2} \equiv h_{\alpha_-,\alpha_\perp}$ as defined by Eq. (30). It is sufficient to look at the expectation value of $: \phi^2 :$ at $x = 0$:

$$\langle : \phi^2(0) : \rangle_{(h_\alpha)^N} = N\langle : \phi^2(0) : \rangle_{h_\alpha} = N\frac{8}{5}\frac{\Gamma(d-2)V_{d-3}}{\pi^{d-2}}\alpha_\perp^{d-2}. \tag{51}$$

This implies a maximum particle number

$$N_{\max} \leq \frac{5}{8}\frac{\pi^{d-2}}{\Gamma(d-2)V_{d-3}}\frac{\phi^2_{\max}}{\alpha_\perp^{d-2}}, \tag{52}$$

for which effective field theory applies. It is easy to see how this cutoff on field values excludes badly behaved negative null energy densities. I.e. let us fix a spacetime neighborhood, $\bar{\mathcal{U}}$,

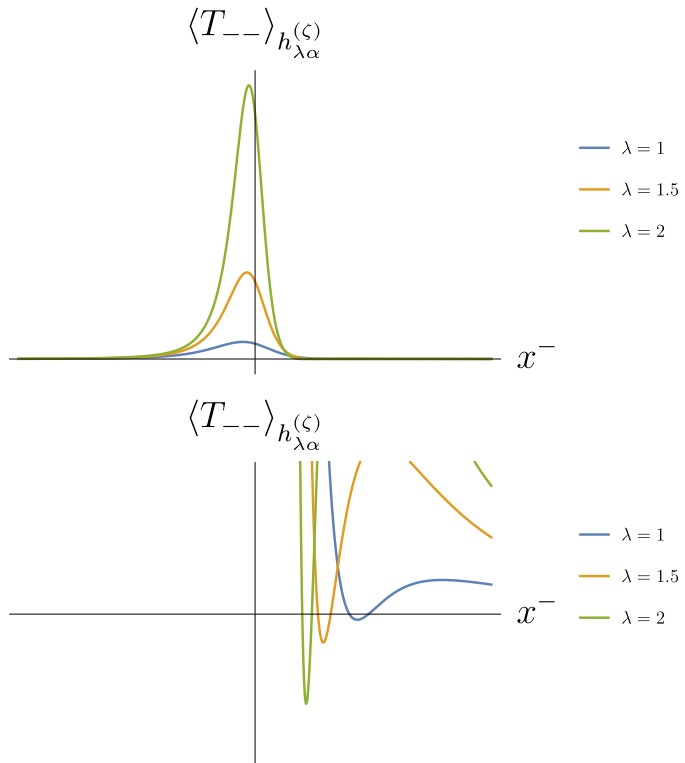

Figure 3: $\langle T_{--}\rangle_{h^{(\zeta)}_{\lambda\alpha}}$ in $n = 4$ dimensions at the point $(x^-, x^+ = \vec{y}_\perp = 0)$ plotted (in units of $\alpha_-^2\alpha_\perp^2$) along $x^-$. The coupling has been set to $\xi = -\frac{1}{200}$. By tuning $\zeta$, in this case to $\zeta = \frac{3}{4}e^{i\frac{\pi}{3}}$, we can find regions of negative null energy density. **(Top)** The null energy plotted in its full range. **(Bottom)** A magnified view illustrating the region of negative null energy density.

about $x = 0$. There is then a set of state parameters $\{\bar{\alpha}_-, \bar{\alpha}_\perp\}$, such that (34) is true. For any $\bar{\rho}$ such that $\langle T_{--}\rangle_{(h_\alpha)^N}$ in this state is upperbounded by $-\bar{\rho}$ for all points in $\bar{\mathcal{U}}$, the combinations of (52) and (46) imply

$$\bar{\rho} < \frac{3}{4}\alpha_-^2\,\xi\,\phi_{\max}^2\,. \tag{53}$$

Furthermore, such a maximum particle number implies that the null energy density is lower-bounded (via (45)) by

$$\langle T_{--}(x)\rangle_{(h_\alpha)^N} \geq -(\mathsf{c}_0 + \mathsf{c}_\xi\xi)\frac{\alpha_-^2}{\pi}\phi_{\max}^2\,, \tag{54}$$

for all points, $x$. Similar conclusions are easily reached for the case of $\xi < 0$ within the general class of $|h^{(\zeta)}_{\alpha_-,\alpha_\perp}\rangle$ states. The upshot of this section is that although we can find a class of states possessing arbitrarily large negative null energy over arbitrarily large regions, such states also display large field values. In particular, restricting to a subclass of states with bounded field values also provides a lower bound on the null energy density in that subclass. Below we will broaden the scope of these indications as well as make them more precise: namely, we will show that for all Hadamard states the null energy density possesses a lower bound that is linked to the field value in that states.

# 4 Null energy inequalities for non-minimal coupling

Building off our general expectation that large negative null energy densities are linked to large field values, in this section we will derive a null energy inequality for the non-minimally coupled scalar. We note that the derivation is for a difference QEIs unlike the absolute one that was possible for the minimally coupled field. Importantly, the strongest bound we will be able to derive is *state-dependent*: i.e. it depends explicitly on the expectation value of $\phi^2$ in the state of interest. This formalizes the expectations we established above in our multi-particle state example. The existence of a state-dependent bound for the null energy matches similar results for the energy density [6] and effective energy density [7]. After the general derivation we discuss the flat-space limit: The inequality of focus here will be the DSNEC introduced in Ref. [15]. Subsequently we will derive the ANEC in this case. Finally we will derive a bound similar to the smeared null energy condition (SNEC) [16].

## 4.1 Quantization

To begin we give a brief introduction to the Hadamard renormalization of free-fields which will be necessary technology for deriving QEIs in curved spacetimes. The quantization procedure will follow the algebraic quantization method described in detail in Ref. [17, 18].

Eq. (8) with $V(\phi) = m^2 \phi^2 / 2$ gives the classical gravitational action for the massive non-minimally coupled free scalar field with Eq. (11) as its equation of motion. To quantize the theory, we introduce an unital $*$-algebra $\mathcal{A}(M)$ on our globally hyperbolic, time-oriented manifold $M$. The algebra is generated by elements of the form $\phi(f)$ where $f$ belongs to the space of compactly supported, smooth, complex-valued functions on $M$, $\mathcal{D}(M)$. Intuitively we can view $\phi(f)$ as a field operator smeared with the test function, $f$. The smeared field operators have to obey the following relations

- Linearity: $\phi(\alpha f + \beta h) = \alpha \phi(f) + \beta \phi(h)$, $\forall\, \alpha, \beta \,\in\, \mathbb{C}$, and $\forall\, f, h \in \mathcal{D}(M)$.

- Hermiticity: $\phi(f)^* = \phi(\bar{f})$, $\forall\, f \in \mathcal{D}(M)$.

- Field equation: $\phi\left( \left( \Box_g + m^2 + \xi R \right) f \right) = 0$, $\forall\, f \,\in\, \mathcal{D}(M)$.

- Canonical Commutation Relations: $[\phi(f), \phi(h)] = iE(f, h)\mathbb{1}$ $\forall\, f, h \in \mathcal{D}(M)$, where $E$ is the difference of the advanced and retarded Green's function for the non-minimally coupled Klein-Gordon operator.

A state of the theory is a linear functional $\omega : \mathcal{A} \to \mathbb{C}$, which is normalized, $\omega(\mathbb{1}) = 1$, and positive $\omega(A^*A) \geq 0$ $\forall A \in \mathcal{A}(M)$. For a given state, $\omega$, the 2-point function is a bi-linear map $W_\omega$ between $\mathcal{D}(M) \times \mathcal{D}(M)$ and $\mathbb{C}$ defined by $W_\omega(f, h) = \omega(\phi(f)\phi(h))$. Not all the states, $\omega$, have physically desirable properties. Additionally, unlike Minkowski space, general spacetimes do not have a preferred vacuum state. Therefore, it is common to choose a class of states known as *Hadamard* states (see Ref. [19] for a review) which have a two-point function singularity structure close to that of the Minkowski vacuum. For instance in $n = 4$ dimensions:

$$W_\omega(x, x') = \frac{U(x, x')}{4\pi^2 \sigma(x, x')} + V(x, x') \log\left( \frac{\sigma}{2\ell^2} \right) + \text{smooth}, \tag{55}$$

where $U$ and $V$ are real and symmetric smooth functions constructed from the metric and the couplings, $\sigma$ is the squared geodesic distance, and $\ell$ is a particular length scale. As a consequence of this definition, the difference between the two-point function of any two Hadamard states is smooth. This property will be used to derive the *difference* QEIs, which provide lower bounds on the expectation value of the averaged stress-energy tensor in a Hadamard state

$\psi$ normal ordered relative to a reference Hadamard state $\psi_0$. Alternatively, we can use the Hadamard parametrix $H_{(k)}(x)$ which encodes the singularity structure of Hadamard states, to derive *absolute* QEIs. $H_{(k)}(x)$ represents the terms up to order $k$ of the infinite Hadamard series. In this case, we do not require a reference state. As this method is not used here we refer the reader to Ref. [20, 21] for more details.

To quantize the stress-energy tensor of the non-minimally coupled scalar field given in Eq. (13), we define the point-split energy operator

$$
\mathbb{T}^{\text{split}}_{\mu\nu}(x, x') = (1 - 2\xi)\nabla^{(x)}_\mu \otimes \nabla^{(x')}_{\nu'} - \frac{1}{2}(1 - 4\xi)\, g_{\mu\nu}(x, x')\Big(\big(m^2 + \xi R\big)\mathbb{1} \otimes \mathbb{1}
$$
$$
- g^{\lambda\rho'}(x, x')\nabla^{(x)}_\lambda \otimes \nabla^{(x')}_{\rho'}\Big) - 2\xi\left(\mathbb{1} \otimes_s \nabla^{(x')}_\mu \nabla^{(x')}_\nu + \frac{1}{2}R_{\mu\nu}\mathbb{1} \otimes \mathbb{1}\right), \qquad (56)
$$

where $\otimes_s$ is the symmetrised tensor product and $g_{\mu\nu}(x, x')$ is the parallel propagator along the unique geodesic connecting $x$ and $x'$. Let's consider a Hadamard state $\psi$ with a two-point function $W_\psi$. To construct the expectation value of the renormalised stress-energy tensor in the state $\psi$ as defined in Ref. [22, 23], $\langle T^{\text{ren}}_{\mu\nu}\rangle_\psi(x)$, we first subtract the Hadamard parametrix to remove the singularities of the two-point function and take the coincidence limit[7]

$$
\langle T^{\text{fin}}_{\mu\nu}\rangle_\psi(x) = \lim_{x' \to x} g_\nu{}^{\nu'}(x, x')\mathbb{T}^{\text{split}}_{\mu\nu'}(W_\psi - H_{(k)})(x, x'). \qquad (57)
$$

It can be shown that $\langle T^{\text{fin}}_{\mu\nu}\rangle_\psi(x)$ is not covariantly conserved, and therefore it is not an appropriate stress-energy tensor. We can recover this property by subtracting a local quantity $Qg_{\mu\nu}$ to $\langle T^{\text{fin}}_{\mu\nu}\rangle_\psi(x)$. Now we impose

$$
\langle T^{\text{ren}}_{\mu\nu}\rangle_\psi - \langle T^{\text{ren}}_{\mu\nu}\rangle_{\psi_0} = \left[\!\!\left[ g_\nu{}^{\nu'}\mathbb{T}^{\text{split}}_{\mu\nu'}(W_\psi - W_{\psi_0})\right]\!\!\right], \qquad (58)
$$

for any two Hadamard states $\psi$ and $\psi_0$. This condition implies that any remaining renormalisation must be a state-independent local curvature term $C_{\mu\nu}$. Finally, we have

$$
\langle T^{\text{ren}}_{\mu\nu}\rangle_\psi(x) = \langle T^{\text{fin}}_{\mu\nu}\rangle_\psi - Q(x)g_{\mu\nu}(x) + C_{\mu\nu}(x). \qquad (59)
$$

The expectation value of the stress tensor in a state $\omega$ normal ordered relative to a reference Hadamard state $\omega_0$

$$
\langle : T_{\mu\nu} : \rangle_\psi := \left[\!\!\left[ g_\nu{}^{\nu'}\mathbb{T}^{\text{split}}_{\mu\nu'}(W_\psi - W_{\psi_0})\right]\!\!\right] = \langle T^{\text{ren}}_{\mu\nu}\rangle_\psi - \langle T^{\text{ren}}_{\mu\nu}\rangle_{\psi_0}, \qquad (60)
$$

allows us to write the difference QEI

$$
\langle :\rho_n:(f)\rangle_\psi = \langle : T_{\mu\nu}\ell^\mu \ell^\nu : (f)\rangle_\psi, \qquad (61)
$$

where $f$ is a test function.

## 4.2 A worldvolume QEI

The null energy of Eq. (14) is written as a quantum field $\phi(f)$ for any test function $f$

$$
\rho_n(f) = T_{\mu\nu}(\ell^\mu \ell^\nu(f)) = (1 - 2\xi)(\nabla_\mu \phi \nabla_\mu \phi)(\ell^\mu \ell^\nu f)
$$
$$
- 2\xi\left((\phi \nabla_{(\mu}\nabla_{\nu)}\phi)(\ell^\mu \ell^\nu f) + \phi^2(R_{\mu\nu}\ell^\mu \ell^\nu f)\right), \qquad (62)
$$

where $T_{\mu\nu}$ is the quantized stress-tensor.

---

[7]We will denote the coincidence limit of a generic bi-distribution, $\mathcal{B}(x, x')$, as $[\![\mathcal{B}(x')]\!] = \lim_{x \to x'} \mathcal{B}(x, x')$.

Leibniz's rule is the 10th axiom of those set by Hollands and Wald [22] for the construction of algebra elements that qualify as local and covariant Wick powers. In our case, we will use the following form

$$\frac{1}{2}(\nabla_\mu\nabla_\nu(\phi^2))(f\ell^\mu\ell^\nu) = (\nabla_\mu\phi\nabla_\nu\phi)(f\ell^\mu\ell^\nu) + (\phi\nabla_{(\mu}\nabla_{\nu)}\phi)(f\ell^\mu\ell^\nu), \tag{63}$$

where the left hand side is understood distributionally so

$$\left(\nabla_\mu\nabla_\nu(\phi^2)\right)(\ell^\mu\ell^\nu f) = \phi^2(\nabla_\mu\nabla_\nu(\ell^\mu\ell^\nu f)). \tag{64}$$

Then Eq. (62) becomes

$$\rho_n(f) = (\nabla_\mu\phi\nabla_\nu\phi)(\ell^\mu\ell^\nu f) - \xi\phi^2\left(\nabla_\mu\nabla_\nu(\ell^\mu\ell^\nu f) + \frac{1}{2}R_{\mu\nu}(\ell^\mu\ell^\nu f)\right). \tag{65}$$

We are interested in expectation values of the quantized null energy in state $\psi$, normal ordered relative to a reference Hadamard state $\psi_0$

$$\langle{:}\rho_n{:}(f)\rangle_\psi = \langle\rho_n(f)\rangle_\psi - \langle\rho_n(f)\rangle_{\psi_0}. \tag{66}$$

Following Ref. [21] we define a small sampling domain $\Sigma$. This is an open subset of $(\mathcal{M}, g)$ that is contained in a globally hyperbolic convex normal neighbourhood of $\mathcal{M}$ and may be covered by a single hyperbolic coordinate chart, $\{x^\mu\}$. The latter requires that $\partial/\partial x^0$ is future pointing and timelike and that there exists a constant $c > 0$ such that

$$c|u_0| \geq \sqrt{\sum_{j=0}^3 u_j^2}, \tag{67}$$

holds for the components of every causal covector, $u$, at each point of $\Sigma$. That statement means that the coordinate speed of light is bounded. Now we may express the hyperbolic chart $\{x^\mu\}$ by a map $\kappa$ where $\Sigma \to \mathbb{R}^n$, $\kappa(p) = (x^0(p), x^1(p), \dots, x^{n-1}(p))$. Any function $g$ on $\Sigma$ determines a function $g_\kappa = g \circ \kappa^{-1}$ on $\Sigma_\kappa = \kappa(\Sigma)$. In particular, the inclusion map $\iota : \Sigma \to \mathcal{M}$ induces a smooth map $\iota_\kappa : \Sigma_\kappa \to \mathcal{M}$. We have $\vartheta : \Sigma \times \Sigma \to \mathcal{M} \times \mathcal{M}$ the map $\vartheta(x, x') = (\iota \otimes \iota)(x, x')$. Here $h = \iota^*g$ is a Lorentzian metric on $\Sigma$ and $h_\kappa$ is the determinant of the matrix $\kappa^*h$. Then the bundle $\mathcal{N}^+$ of non-zero future pointing null covectors on $(\mathcal{M}, g)$ pulls back under $\iota_\kappa$ so that

$$\iota_\kappa^*\mathcal{N}^+ \subset \Sigma_\kappa \times \mathcal{D}, \tag{68}$$

where $\mathcal{D} \subset \mathbb{R}^n$ is the set of all $u_a$ that satisfy Eq. (67).

Now let $f$ be any real-valued test function compactly supported in the small sampling domain $\Sigma$. Let the Hadamard reference state $\psi_0$ have a 2-point function, $W_0$. Using Eq. (65) the expectation values of the null energy density in Hadamard state $\psi$ and normal-ordered relative to $\psi_0$, can be written as

$$\int_\Sigma d\mathrm{vol}(x) f^2(x)\langle{:}T_{\mu\nu}\ell^\mu\ell^\nu{:}\rangle_\psi = \int_\Sigma d\mathrm{vol} f(x)^2[\![\hat{\rho}_n{:}W_\psi{:}]\!] - \xi\langle{:}\phi^2{:}(\mathfrak{Q}[f])\rangle_\psi, \tag{69}$$

where

$$\hat{\rho}_n = \ell^\mu\nabla_\mu \otimes \ell^\nu\nabla_\nu, \tag{70a}$$

$$\mathfrak{Q}[f] = \nabla_\mu\nabla_\nu(\ell^\mu\ell^\nu f^2) + \frac{1}{2}R_{\mu\nu}\ell^\mu\ell^\nu f^2. \tag{70b}$$

The operator $\hat{\rho}_n$ is symmetric and positive definite so we can use the following inequality to bound it [7, 21]

$$\int_\Sigma d\mathrm{vol} f^2(x) [\![ Q \otimes Q {:} W {:} ]\!] \geq -2 \int_\mathcal{D} \frac{d^n\alpha}{(2\pi)^n}((Q \otimes Q)W_0)_\kappa(\bar{f}_\alpha, f_\alpha) > -\infty, \tag{71}$$

where

$$f_\alpha(x) = e^{i\alpha x} f(x). \tag{72}$$

Then we can state the following theorem

**Theorem 4.1.** *Let $f$ be any real-valued test function compactly supported in a small sampling domain. Let $\ell^\mu$ be a null vector field defined in the neighbourhood of support of $f$. Then, for all Hadamard states $\psi$*

$$\langle :\rho_n : (f^2) \rangle_\psi \geq -2 \int_\mathcal{D} \frac{d^n\alpha}{(2\pi)^n}((Q \otimes Q)W_0)_\kappa(\bar{f}_\alpha, f_\alpha) - \xi\langle :\phi^2 : (\mathfrak{Q}[f]) \rangle_\psi, \tag{73}$$

*where $W_0$ is the two-point function of the reference Hadamard state $\psi_0$ and $Q = \ell^\mu \nabla_\mu$.*

This QEI precisely expresses the connection between negative null-energy densities and large fluctuating field values. Namely: The state-dependence of the QEI indicates the possibility of finding states with arbitrarily negative null-energy densities on the small sampling domain, but only at the expense of having a largely fluctuating scalar field. The above QEI is similar in nature with the derived QEIs for the non-minimally coupled scalar for the energy density [6] and the effective energy density [7]. One important difference is that unlike those two inequalities, the QEI derived holds for any value of the coupling constant $\xi$.

We emphasize at this point that the QEI of Eq.(73) holds for any curved manifold assuming that we have a known reference state with two-point function $W_0$. While there is no lower bound for unbounded $\langle :\phi^2 : \rangle_\psi$, adopting the EFT framework Eq.(73) bounds the null-energy from below by the maximum value of the Wick square. The ANEC could also be derived for a curved background with bounded curvature following the methods of [24] and [25]. Thus one of the main results of this work holds for general curvature.[8]

We should also note that this bound is non-trivial. Let's assume a general state-dependent QEI of the form

$$\langle :\rho : (f^2) \rangle_\psi \geq -\langle \mathfrak{Q}(f) \rangle_\psi, \tag{74}$$

where $\rho$ is some contraction of the stress-energy tensor and $\mathfrak{Q}(f)$ an operator which is allowed to be unbounded. A trivial bound would be one where, for example $\mathfrak{Q}(f) = -\rho(f)$. In Ref. [6] it was shown that a state-dependent bound is non trivial if there exist constants $c$ and $c'$ such that

$$\langle \rho(f) \rangle_\psi \geq c + c' \left| \langle \mathfrak{Q}(f) \rangle_\psi \right|, \tag{75}$$

for all states $\psi$ in the class unless $f$ is identically zero. As it was shown in Refs. [6] and [7] a bound where the state dependence is isolated in the form of the Wick square the bound is non-trivial. Thus the bound of Eq. (73) is a non-trivial bound.

Finally, as this bound is derived by discarding positive terms we do not expect it to be optimal. There are no examples in more than two dimensions where the QEI bounds are saturated.

---

[8]The details of the curvature corrections and the covariance of the DSNEC are important and the topic of future work.

### 4.3 DSNEC and ANEC

Our above result, (73), applies generally for non-minimally coupled scalar fields in a curved background smeared over a smooth sampling domain. It is also useful for us to discuss this bound in $n$-dimensional Minkowski spacetime and make comparisons to the DSNEC proven in [15]. As discussed above, we can equivalently view this as a bound on "improved stress tensors" in a free massive scalar field theory.

We will, as usual in this context, renormalize with the Minkowski vacuum state as the reference Hadamard state. Note that while the expression of the null energy operator remains unchanged with the introduction of a mass, the vacuum state $\psi_0$ changes from massless to massive Minkowski vacuum. We have

$$\int_\Sigma d\mathrm{vol}(x) f^2(x) \langle :T_{\mu\nu}\ell^\mu\ell^\nu: \rangle_\psi \geq -2 \int_{\mathcal{D}} \frac{d^n\alpha}{(2\pi)^n} ((Q\otimes Q)W_0)_\kappa(\bar{f}_\alpha, f_\alpha)$$
$$-\xi \int_\Sigma d\mathrm{vol}(x) \langle :\phi^2: \rangle_\psi \nabla_\mu \nabla_\nu (\ell^\mu \ell^\nu f^2). \tag{76}$$

To derive DSNEC we want to restrict the domain $\Sigma$ to the two null dimensions $x^\pm$. To do so we write the smearing function as

$$f(x^+, x^-, \vec{y})^2 = f(x^+, x^-)^2 \delta^{n-2}(\vec{y}). \tag{77}$$

Using the results of Ref. [15], Eq. (76) becomes

$$\int d^2x^\pm f(x^\pm)^2 \langle :T_{--}: \rangle_\psi \geq -\frac{8}{\pi^{n/2-2}\Gamma\left(\frac{n-2}{2}\right)} \int_{\mathcal{D}} \frac{d^2\alpha_\pm}{(2\pi)^2} \int \frac{d^2k_\pm}{(2\pi)^2} |\tilde{f}(k_\pm)|^2$$
$$\times \zeta_-^2 \left(4\zeta_+\zeta_- - m^2\right)^{\frac{n-4}{2}} \Theta\left(4\zeta_+\zeta_- - m^2\right)\Theta(\zeta_-)\Big|_{\zeta_\pm=k_\pm-\alpha_\pm}$$
$$-\xi \int d^2x^\pm \langle :\phi^2: \rangle_\psi \partial_-^2 (f(x^\pm)^2). \tag{78}$$

Now we restrict the domain $\mathcal{D}$ to the boosted domains $\mathcal{D}_\eta = \{\alpha_\eta = e^\eta\alpha_+ + e^{-\eta}\alpha_- \geq 0\}$. Then we have

$$\int d^2x^\pm f(x^\pm)^2 \langle :T_{--}: \rangle_\psi \geq -\frac{e^{2\eta}(4\pi)^{\frac{1-n}{2}}}{\Gamma\left(\frac{n+1}{2}\right)} \int \frac{d^2k_\pm}{(2\pi)^2} |\tilde{f}(k_\pm)|^2 k_\eta (k_\eta^2 - m^2)^{\frac{n-1}{2}} \Theta(k_\eta - m)$$
$$-\xi \int d^2x^\pm \langle :\phi^2: \rangle_\psi \partial_-^2 (f(x^\pm)^2), \tag{79}$$

where $k_\eta = e^\eta k_+ + e^{-\eta}k_-$. At this point, the state-dependent term, $\langle :\phi^2: \rangle_\psi$, prevents one from writing down a DSNEC in its standard form à la [15]. This is expected: we have already seen in section 3 there exist states which can violate such smeared inequalities. As emphasized in that section, such states also have large field values. We can make progress if we focus on the class of states obeying

$$\left|\langle :\phi^2: \rangle_\psi\right| \leq \phi_{\max}^2, \tag{80}$$

where $\phi_{\max}$ is a finite constant. We will motivate such a bound below in section 5 by regarding the such states as part of an effective field theory. In this case, it is immediate that we can write Eq. (79) as

$$\int d^2x^\pm f(x^\pm)^2 \langle :T_{--}: \rangle_\psi \geq -\min_{\eta\in\mathbb{R}} \frac{e^{2\eta}(4\pi)^{\frac{1-n}{2}}}{\Gamma\left(\frac{n+1}{2}\right)} \int \frac{d^2k_\pm}{(2\pi)^2} |\tilde{f}(k_\pm)|^2 k_\eta (k_\eta^2 - m^2)^{\frac{n-1}{2}} \Theta(k_\eta - m)$$
$$-|\xi|\phi_{\max}^2 \int d^2x^\pm \left|\partial_-^2 (f(x^\pm)^2)\right|. \tag{81}$$

For the massless case, and when the test function factorizes as

$$|\tilde{f}(k_+, k_-)|^2 = |\tilde{f}(k_+)|^2 |\tilde{f}_-(k_-)|^2,\tag{82}$$

we can perform an $\eta$ optimization [15] which gives

$$\int d^2x^\pm f(x^\pm)^2 \langle :T_{--}: \rangle_\psi \geq -P_n \langle |k_+|^n \rangle^{\frac{n-2}{2n}} \langle |k_-|^n \rangle^{\frac{n+2}{2n}}$$

$$- |\xi|\phi_{\max}^2 \int d^2x^\pm |\partial_-^2(f(x^\pm)^2)|,\tag{83}$$

where $\langle |k_\pm|^p \rangle := \int d^p k_\pm |\tilde{f}_\pm|^2 |k_\pm|^p$ are the moments and $P_n$ is a constant that only depends on the number of spacetime dimensions. In the case of even dimensions, the integrals can be inverse Fourier transformed to integrals in position space and we have

$$\int d^2x^\pm f(x^\pm)^2 \langle :T_{--}: \rangle_\psi \geq -P_n \left( \int dx^+ (f_+^{(n/2)}(x^+))^2 \right)^{\frac{n-2}{2n}} \left( \int dx^- (f_-^{(n/2)}(x^-))^2 \right)^{\frac{n+2}{2n}}$$

$$- |\xi|\phi_{\max}^2 \int d^2x^\pm |\partial_-^2(f(x^\pm)^2)|.\tag{84}$$

Starting out from the massive case of Eq. (81) we can scale out the support of the smearing function $f$ in the two null directions, $\delta^\pm$

$$f(x^+, x^-) = \frac{1}{\sqrt{\delta^+ \delta^-}} \mathcal{F}(x^+/\delta^+, x^-/\delta^-),\tag{85}$$

where $\mathcal{F}(s^+, s^-)$ is a function of dimensionless variables dropping off quickly for $|s^\pm| \gg 1$ and normalized to $\int d^2s^\pm \mathcal{F}(s^+, s^-)^2 = 1$. We can write Eq. (79) as

$$\int \frac{d^2x^\pm}{\delta^+\delta^-} \mathcal{F}(x^+/\delta^+, x^-/\delta^-)^2 \langle T_{--} \rangle_\psi \geq -\frac{\mathcal{N}_n[\gamma]}{(\delta^+)^{\frac{n-2}{2}}(\delta^-)^{\frac{n+2}{2}}} - \frac{\mathcal{C}}{(\delta^-)^2}|\xi|\phi_{\max}^2,\tag{86}$$

where $\mathcal{C}$ is a numerical factor depending on the smearing and $\mathcal{N}_n$ is dimensionless function of the dimensionless mass, $\gamma^2 := \delta^+\delta^- m^2$ (see [15] for an explicit expression) or

$$\int \frac{d^2x^\pm}{\delta^+\delta^-} \mathcal{F}(x^+/\delta^+, x^-/\delta^-)^2 \langle T_{--} \rangle_\psi \geq -\frac{\tilde{\mathcal{N}}_n[\gamma, \xi, \tilde{\phi}_{\max}^2]}{(\delta^+)^{\frac{n-2}{2}}(\delta^-)^{\frac{n+2}{2}}}.\tag{87}$$

Above $\tilde{\mathcal{N}}_n$ is a new function including a dependence on the dimensionless maximum field value, $\tilde{\phi}_{\max}^2 = (\delta^+\delta^-)^{\frac{n-2}{2}} \phi_{\max}^2$, and is given by the minimization of the integrals

$$\tilde{\mathcal{N}}_n = \min_{\tilde{\eta} \in \mathbb{R}} \frac{e^{2\tilde{\eta}}}{(4\pi)^{\frac{n-1}{2}}\Gamma\left(\frac{n+1}{2}\right)} \int \frac{d^2\rho_\pm}{(2\pi)^2} |\tilde{\mathcal{F}}(\rho_+, \rho_-)|^2 \rho_{\tilde{\eta}}(\rho_{\tilde{\eta}}^2 - \gamma^2)^{\frac{n-1}{2}} \Theta(\rho_{\tilde{\eta}} - \gamma)$$

$$+ |\xi|\tilde{\phi}_{\max}^2 \int d^2s^\pm |\partial_{s^-}^2(\mathcal{F}(s^\pm)^2)|,\tag{88}$$

where we've denoted $e^{\tilde{\eta}} := \sqrt{\frac{\delta^-}{\delta^+}} e^\eta$ and $\rho_\pm := \delta^\pm k_\pm$ as well as $\rho_{\tilde{\eta}} = e^{\tilde{\eta}} \rho_+ + e^{-\tilde{\eta}} \rho_-$. Lastly, $\tilde{\mathcal{F}}(\rho_\pm) := \int d^2s\, e^{i\rho_\pm s^\pm} \mathcal{F}(s^\pm)$ is the dimensionless Fourier-transform of $\mathcal{F}$.

Now we can show that using Eq. (87) we recover ANEC. We want to take the limit $\delta^+ \to 0$ and $\delta^- \to \infty$ while holding $\delta^+\delta^- \equiv \alpha^2$ fixed. To recover the ANEC limit we require that the smearing function satisfies

$$\lim_{\delta^+ \to 0} \lim_{\delta^- \to \infty} \frac{1}{\delta^+} \mathcal{F}(x^+/\delta^+, x^-/\delta^-)^2 = A\delta(x^+ - \beta),\tag{89}$$

where $A$ and $\beta$ are real numbers. Then the first term of Eq.(87) goes as $\delta^+/(A\alpha^n)$ while the second term goes as $\delta^+/(A\alpha^4)$. Both terms go to zero as $\delta^+ \to 0$. So we have

$$\int_{-\infty}^{\infty} dx^- \langle T_{--}(x^+ = \beta, x^-, \vec{y} = 0)\rangle_\psi \geq 0\,. \tag{90}$$

We see that, as in the classical case discussed in Sec. 2.2 we recover ANEC for any coupling in Minkowski spacetime. In the classical case we can do that, for bounded field values, for any spacetime curvature using the effective null energy.

## 4.4 SNEC

We can additionally show that our bound implies a form of SNEC [16]. We impose the following cutoff: we take $\delta^+ \to 0$ while $\delta^+\delta^- \to \ell_{UV}^2$. We require that the smearing function factorizes, $f(x^+, x^-) = \left(\mathcal{F}_+(x^+/\delta^+)/\sqrt{\delta^+}\right)f(x^-)$ and

$$\lim_{\delta^+ \to 0} \mathcal{F}_+(x^+/\delta^+)^2/\delta_+ = \delta(x^+ - \beta) = \frac{1}{\delta_+}\delta(x^+/\delta_+ - \beta/\delta_+)\,. \tag{91}$$

Then

$$\int dx^- f_-(x^-)^2 \langle T_{--}(x^+ = \beta, x^-)\rangle_\psi \geq -\frac{\mathcal{N}_n[\gamma, \xi, \tilde{\phi}_{\max}^2]}{\ell_{UV}^{n-2}(\delta^-)^2}\,. \tag{92}$$

In Ref. [26] the SNEC for the minimally coupled scalar was used to prove a Penrose-type singularity theorem. This can be done directly using the methods of Ref. [27] if the energy condition is of the form

$$\int dx^- f_-(x^-)^2 \langle T_{--}\rangle_\psi \geq -Q_m\|f_-^{(m)}\|^2 - Q_0\|f_-\|^2\,, \tag{93}$$

where $\|\cdot\|^2$ is the $L_2$ norm and $m$ is a positive integer that denotes the number of derivatives. Here $Q_m$ and $Q_0$ are non-negative constants.

To derive such a bound we specialize to the case of the massless scalar field where we can use Eq. (83). To take the first term in the SNEC limit we can either bound the $\pm$ momenta independently or impose a covariant cutoff $(k_-)_{\max}(k_+)_{\max} \ll \ell_{UV}^{-2}$ as described in Ref. [15]. Both assumptions lead to the same bound. For the second term of Eq. (83), the SNEC limit is straightforward so we have

$$\int dx^- f_-(x^-)^2 \langle T_{--}\rangle_\psi \geq -\frac{p_n}{\ell_{UV}^{n-2}} \int dx^- (\partial_- f(x^-))^2 - \frac{|\xi|\tilde{\phi}_{\max}^2}{\ell_{UV}^{n-2}} \int dx^- \left|(\partial_-^2(f(x^-)^2))\right|\,, \tag{94}$$

where we used the fact that the smearing function factorizes and the limit of Eq. (91). Here $p_n = 4(n-2)^{-1}\pi^{-n/2}\Gamma((n-2)/2)^{-1}$. To write the bound in the form of Eq. (93) we use the inequality

$$\int dx^- (f'_-)^2 \leq \frac{1}{2\ell_{UV}^2}\|f_-\|^2 + \frac{\ell_{UV}^2}{2}\|f''_-\|^2\,. \tag{95}$$

Then we have

$$\int dx^- f_-(x^-)^2 \langle T_{--}\rangle_\psi \geq -Q_2\|f_-^{(2)}\|^2 - Q_0\|f_-\|^2\,, \tag{96}$$

where

$$Q_0 = \frac{1}{\ell_{UV}^n}\left(\frac{p_n}{2} + 2|\xi|\tilde{\phi}_{\max}^2\right), \qquad Q_2 = \frac{1}{\ell_{UV}^{n-4}}\left(\frac{p_n}{2} + 2|\xi|\tilde{\phi}_{\max}^2\right)\,. \tag{97}$$

Here we also used the triangle inequality and the classical inequality

$$ab \leq \varepsilon a^2 + \frac{1}{4\varepsilon}b^2\,,\tag{98}$$

for $a = |f|$, $b = |f''|$ and $\varepsilon := (2\ell_{\text{UV}}^2)^{-1}$.

Then the results of Ref. [26] that proved a singularity theorem and of Ref. [28] that proved the generalized area theorem using SNEC, can be applied to the non-minimally coupled theory with modified coefficients. If we take the EFT approach as before, and consider $\phi_{\text{max}}^2 \sim M_{\text{cutoff}}^{n-2} \sim \ell_{\text{UV}}^{-(n-2)}$, then $\tilde{\phi}_{\text{max}}$ is of order 1. Of course a different bound might be imposed for $\phi_{\text{max}}$ as in Ref. [29].

## 5 Non-minimal coupling as an effective field theory

Above we have shown that the non-minimally coupled scalar theory has a null-energy density which is bounded below by a state-dependent term: The smeared value of the scalar field. In particular, this implies that smeared null-energy can be arbitrarily negative but only in states with large scalar field two-point functions. For the class of states obeying $\left|\langle : \phi^2 : \rangle_\psi\right| < \phi_{\text{max}}^2$ we can derive a state-independent QEI that admits a DSNEC form and has the potential to be utilized in a singularity theorem. We interpret this statement as the suggestion that the non-minimally coupled theory should be regarded as an effective field theory (EFT) of states with not only bounded momenta but also bounded scalar field values. In this section we further this argument presenting several points of evidence for this interpretation and suggest that $\phi_{\text{max}}^2 \lesssim (8\pi G_N|\xi|)^{-1}$.

We will present arguments in two complementary manners. In Sec. 2.3 we discussed the conformal transformation and field redefinition that connects the Einstein and Jordan frames. At the classical level, the stress tensors of these frames are different: The Jordan frame stress tensor is NEC violating while the Einstein frame stress tensor is NEC satisfying. Which operator is relevant is depends on which frame one considers physical. However, classically the two stress tensors are related by a straight-forward map. The above statements remain sensible in the quantum theory when interpreted as an EFT with bounded field values. Namely, the stress tensors of two frames, while different quantum operators, as easily related and with controlled quantum corrections. Contrastingly, we will argue that the quantum theory experiences breakdowns in its perturbative and semi-classical control in both frames when we allow for large field values. In the Einstein frame this breakdown happens in the scalar sector through a tower of irrelevant interactions that become important when $\phi^2 \sim (8\pi G_N\xi)^{-1}$; in the Jordan frame it occurs in the gravitational sector where the gravity path-integral becomes strongly coupled and looses its saddle-point approximation.

### 5.1 The Einstein frame

Let us begin with change of frame at the level of the action. These manipulations are valid in the classical or quantum theory, when viewed as taking place within the path-integral. Later in this section, we will consider additional corrections that arise in the quantum theory due to change in path integral measure.

The conformal transformation and field redefinition

$$\tilde{g}_{\mu\nu} = \Omega^2(\phi(x))g_{\mu\nu}\,,\qquad \tilde{\phi}(x) = F(\phi(x))\,,\tag{99}$$

bringing the non-minimally coupled action to (23) is given by

$$\Omega(\phi) = \left(1 - 8\pi G_N\xi\phi^2\right)^{\frac{1}{n-2}}\,,\tag{100}$$

with $F$ satisfying

$$F'(\phi) = \left(1 - 8\pi G_N \xi \, \phi^2\right)^{-1} \sqrt{1 - 8\pi G_N \, \xi \left(1 - \frac{\xi}{\xi_c}\right)\phi^2} \,, \tag{101}$$

where $\xi_c$ is the conformal coupling. This leads to an effective potential given by

$$\tilde{V}(\tilde{\phi}) = \left(1 - 8\pi G_N \, \xi \, \phi^2\right)^{\frac{n}{2-n}} \left(\frac{\Lambda}{8\pi G_N} + V(\phi)\right), \quad \phi = F^{-1}(\tilde{\phi}). \tag{102}$$

This mapping is possible for a generic class of curvature coupings of the form $\mathcal{A}[\phi]R$ (see Appendix A) however above we have written it explicitly for the simplest non-minimal coupling, (8), which can be seen as the first term in a derivative and polynomial expansion in the effective field theory for $\phi$. Classically, the Einstein frame stress-tensor is

$$\tilde{T}_{\mu\nu}^{\text{classical}} = (\tilde{\nabla}_\mu \tilde{\phi})(\tilde{\nabla}_\nu \tilde{\phi}) - \frac{1}{2}\tilde{g}_{\mu\nu}(2\tilde{V}(\tilde{\phi}) + (\tilde{\nabla}\tilde{\phi})^2). \tag{103}$$

The potential is generally non-polynomial in $\tilde{\phi}$. To explicitly illustrate this, let us take the free conformally coupled scalar ($\xi = \xi_c$), where we can solve (101) exactly:

$$F(\phi) = \frac{\text{arctanh}\left(\sqrt{8\pi G_N \xi_c}\phi\right)}{\sqrt{8\pi G_N \xi_c}} \,. \tag{104}$$

In $n = 4$ dimensions the effective potential is then

$$\tilde{V}(\tilde{\phi}) = \frac{\Lambda}{8\pi G_N}\cosh^4\left(\sqrt{8\pi G_N \xi}\tilde{\phi}\right) + \frac{m^2}{16\pi G_N \xi}\sinh^2\left(\sqrt{32\pi G_N \xi}\tilde{\phi}\right). \tag{105}$$

More generally, for arbitrary $\xi$, we must proceed through a power series expansion:

$$\tilde{\phi} = \phi\left(1 + \frac{1}{6}\left(1 + \frac{\xi}{\xi_c}\right)(8\pi G_N \xi \phi^2) + \dots\right), \tag{106}$$

leading to an effective potential

$$\tilde{V}(\tilde{\phi}) = \frac{\Lambda}{8\pi G_N} + \frac{1}{2}\left(m^2 + 4\xi \Lambda\right)\tilde{\phi}^2 + \frac{1}{6}\left(m^2\left(5 - \frac{\xi}{\xi_c}\right) + 2\Lambda\xi\left(7 - 2\frac{\xi}{\xi_c}\right)\right)(8\pi G_N \xi)\tilde{\phi}^4 + \dots, \tag{107}$$

in $n = 4$ dimensions. The perturbative parameter in this expansion is $8\pi G_N \xi \tilde{\phi}^2$. To first order in this perturbative parameter, this is a massive theory with a quartic interaction, $\frac{\lambda}{4!}\tilde{\phi}^4$, with an effective mass and coupling

$$m_{\text{eff}}^2 = m^2 + 4\xi \Lambda, \qquad \lambda = 4\left(m^2\left(5 - \frac{\xi}{\xi_c}\right) + 2\Lambda\xi\left(7 - 2\frac{\xi}{\xi_c}\right)\right)(8\pi G_N \xi), \tag{108}$$

respectively.

Regardless of the expansion of $\tilde{V}$, the classical Einstein frame stress-tensor, $T_{\mu\nu}^{\text{classical}}$, will obey the NEC. A theorem states that for free bosonic theories if the classical theory obeys a pointwise energy condition then the quantum theory obeys the respective QEI with a state-independent bound [30, 31]. Despite obeying the NEC, the classical Einstein frame theory is self-interacting and thus evades the above theorem. We shouldn't expect to have a state-independent null QEI in this case.

**Quantum corrections**

We now consider the path-integral itself where we will argue within this EFT framework that the field redefinition remains sensible. We will work in Euclidean signature and write the path-integral of gravity coupled to matter schematically[9] as

$$
Z_{\text{grav+matter}} = \int \mathcal{D}g_{\mu\nu}\mathcal{D}\phi\, e^{-I_\xi[\phi,g,V]}, \tag{109}
$$

where

$$
I_\xi[\phi,g,V] = \int d^n x \sqrt{g}\left(-\frac{1}{16\pi G_N}(R-2\Lambda) + \frac{1}{2}(\nabla\phi)^2 + \frac{1}{2}m^2\phi^2 + \frac{\xi}{2}R\phi^2 + V(\phi)\right). \tag{110}
$$

Inside the path-integral we realize the field redefinitions, Eq. (22), as a change of path-integral variables $(g_{\mu\nu},\phi) \rightarrow (\tilde{g}_{\mu\nu},\tilde{\phi})$. As we have seen above the action changes to a minimally coupled action ($\xi \rightarrow \tilde{\xi} = 0$) by design but with a new potential, $\tilde{V}$:

$$
Z_{\text{grav+matter}} = \int \mathcal{D}\tilde{g}_{\mu\nu}\mathcal{D}\tilde{\phi}\, e^{-I_0[\tilde{\phi},\tilde{g},\tilde{V}]+\log J[\tilde{\phi},\tilde{g}]}. \tag{111}
$$

Importantly we also generate a Jacobian given by the functional determinant

$$
J[\tilde{\phi},\tilde{g}] = \det\begin{pmatrix} \frac{\delta g_{\mu\nu}}{\delta\tilde{g}_{\mu\nu}} & \frac{\delta g_{\mu\nu}}{\delta\tilde{\phi}} \\ 0 & \frac{\delta\phi}{\delta\tilde{\phi}} \end{pmatrix} = \det\frac{\delta g_{\mu\nu}}{\delta\tilde{g}_{\mu\nu}}\det\frac{\delta\phi}{\delta\tilde{\phi}}. \tag{112}
$$

Calculating $J$ explicitly is sensitive to the regulation scheme; an explicit computation of the Jacobian would be necessary to make the following details exact but this is beyond the scope of this work.[10] We can instead discuss schematically what terms could appear. If (apart from the volume form) $J$ is simply a function of $\tilde{\phi}$, $\log J[\tilde{\phi},\tilde{g}] = \int d^d x \sqrt{\tilde{g}}\log\hat{J}[\tilde{\phi}]$ then this Jacobian can be absorbed into $\tilde{V}[\tilde{\phi}]$ which we can treat perturbatively as we did above, Eq. (107). If the field redefinition were simply $\phi \rightarrow \tilde{\phi}$ this would be the expectation since the field redefinition does not explicitly involve the metric. In this case, we might be able to treat the Einstein frame as a perturbative $\lambda\tilde{\phi}^4$ theory when the coupling constant is small.

However we also need to consider the redefinition of the metric. More generally, we should expect $J$ itself will induce additional non-minimal scalar-metric couplings, $\mathsf{F}_1(\tilde{\phi})\tilde{R}$, $\mathsf{F}_2(\tilde{\phi})\tilde{R}^2,\ldots$, the leading term of which is the familiar $\tilde{\phi}^2\tilde{R}$. Dimensional analysis suggests that higher derivative, higher polynomial, and higher curvature terms will be controlled by powers of $M_{\text{cutoff}}^{-1}$, the cutoff used in a regularization scheme to compute $J$. The natural scale for this computation is $M_{\text{cutoff}} \sim M_{\text{Planck}}$. Although $\tilde{\phi}^2\tilde{R}$ is possibly generated by a Jacobian, we expect that through successive field redefinitions one can arrive at a theory with minimal coupling. The expense of this is generating (and suitably redefining) *all possible irrelevant couplings*. This is simply stating that in an EFT one should be able to make the leading kinetic terms (for both the metric and the scalar field) canonical as long as one keeps all other possible couplings. This EFT remains valid when the irrelevant couplings are suppressed by $M_{\text{cutoff}}^{-1}$ and puts a natural upper bound on $\xi\tilde{\phi}^2$ in states described by the theory.

We would be remiss if we did not mention the possibility of $\phi$ appearing as a psuedo-Nambu-Goldstone mode of a spontaneously broken, approximate symmetry, which has been much discussed in effective field theories of slow-roll inflation [33, 34]. When $\phi$ enjoys a shift symmetry that is weakly broken, polynomial contributions to its effective potential are

---

[9]It should be understood that the path-integral measure, $\mathcal{D}g_{\mu\nu}$, includes the quotient over gauge-orbits.

[10]However see [32] for an example calculation in a similar theory.

naturally controlled by the scale of weakly broken shift symmetry.[11] This is already evident in Eq. (107) where the quartic term comes equipped with $m^2$ and $\Lambda$, which control the broken shift symmetry of $\phi$ in the Jordan frame. If these scales remain parametrically small this provides a technically natural mechanism for controlling the polynomial contributions to both $\tilde{V}$ and $\log J$ while allowing super-Planckian field values.

We do not disallow this possibility. Instead we offer the more conservative conclusion: when viewed as an EFT with field values bounded by $\left|8\pi G_N \xi \phi^2\right| \ll 1$, the mapping from Jordan frame to Einstein frame can also be done in the quantum theory. Quantum corrections arising from the path integral measure may modify the explicit form of the Einstein frame potential, but its expansion remains controlled.

## 5.2   The Jordan frame

We have argued above that moving from the Jordan frame to the Einstein frame is sensible in an EFT where $\left|8\pi G_N \xi \phi^2\right| \ll 1$. Otherwise potential trouble arises in the scalar sector of the Einstein frame theory as a tower of unsuppressed irrelevant interactions. Here we ask if an analogous breakdown in the Euclidean path-integral in the original Jordan frame.

For positive $\xi$ and fixed, positive, $R$, non-minimal coupling seems fairly innocuous: it behaves as an effective mass. Focussing on the scalar sector, it is then hard to see what trouble arises for large field values. However what we will show below is that large field values in the Jordan frame leads to a breakdown of the semi-classical approximation in the gravitational sector. To be clear about the scope of this section, our following arguments will apply for the $\xi > 0$; we will finish with speculative comments for the case $\xi < 0$.

To begin this discussion we will need to introduce some technology. Namely we would like to explore the effect of changing the field value while also treating the gravitational and scalar path-integrals semi-classically. However finding true saddle-point solutions with a non-zero scalar background is not possible. Instead we will look for solutions subject to the constraint of a non-zero scalar field. While not true saddles, these can be treated in the framework of *constrained instantons* [36] in the following way.[12] Inside the Euclidean path-integral, (109), we insert "one" in the form of a delta function constraint and an integration over the constraint:

$$Z_{\text{grav+matter}} = \int \mathcal{D}g_{\mu\nu}\mathcal{D}\phi \int d(\bar{\phi}^2) \int d\lambda\, e^{-\frac{1}{2}\lambda \int d^n x \sqrt{g}(\phi^2-\bar{\phi}^2)} e^{-I_\xi[\phi,g,V=0]}. \tag{113}$$

The constraint in question, given above as an integral of $\lambda$ parallel to imaginary axis, enforces that the zero mode of $\phi^2$ equals $\bar{\phi}^2$ which we take to be constant.[13] By subsequently integrating over $\bar{\phi}^2$ we can collapse the constraint to obtain the original path-integral. Alternatively, however, we can pull the $\bar{\phi}^2$ integral to the outside and view this as an integral over constrained theories:

$$Z_{\text{grav+matter}} = \int d(\bar{\phi}^2) Z_{\text{con}}[\bar{\phi}^2], \qquad Z_{\text{con}} = \int \mathcal{D}g_{\mu\nu}\mathcal{D}\phi\, d\lambda\, e^{-\frac{1}{2}\lambda \int d^n x \sqrt{g}(\phi^2-\bar{\phi}^2)-I_\xi[\phi,g,V=0]}, \tag{114}$$

and look for saddles of $Z_{\text{con}}$ at fixed $\bar{\phi}^2$. The saddle-point equations for $g_{\mu\nu}$, $\phi$, and $\lambda$ are,

---

[11]For a friendly and comprehensive review of models of inflation and their realizations in string theory see [35] and references there-in.

[12]For other applications of constrained saddles to cosmological spacetimes see [37–39].

[13]In order for this to be a sensible modulus over which to integrate, it will be necessary to give the theory an initial positive cosmological constant (which could be small) and take the boundary conditions on the gravity path-integral to be over compact manifolds with finite Euclidean volume. In practice, we will see that the mass will induce positive background curvature in the constrained saddles.

respectively,

$$(1 - 8\pi G_N \xi \phi^2)G_{\mu\nu} + \Lambda g_{\mu\nu} - 8\pi G_N T^{(\lambda)}_{\mu\nu} = 0,$$
$$\nabla^2 \phi + (\lambda + m^2 + \xi R)\phi = 0,$$
$$\int d^n x \sqrt{g}\left(\phi^2 - \bar{\phi}^2\right) = 0, \tag{115}$$

with

$$T^{(\lambda)}_{\mu\nu} = \nabla_\mu \phi \nabla_\nu \phi - \frac{1}{2} g_{\mu\nu}\left((\nabla\phi)^2 + m^2\phi^2 + \lambda\phi^2 - \lambda\bar{\phi}^2\right). \tag{116}$$

We find a saddle-point solution[14] with a constant $\phi = \bar{\phi}$ background and constant $\bar{R}$ as

$$\bar{R}_{\mu\nu} = \frac{2}{n-2} \frac{\Lambda + 4\pi G_N m^2 \bar{\phi}^2}{1 - 8\pi G_N \xi \bar{\phi}^2} \bar{g}_{\mu\nu},$$
$$\bar{\lambda} = -m^2 - \xi\bar{R}, \tag{117}$$

where the trace of the Einstein equation was used. That is, the saddle-point Euclidean spacetime is a homogeneous space whose radius of curvature is

$$\ell^2_{\text{eff}} = \frac{(n-2)(n-1)}{2} \frac{\left|1 - 8\pi G_N \xi \bar{\phi}^2\right|}{\left|\Lambda + 4\pi G_N m^2 \bar{\phi}^2\right|}, \tag{118}$$

and whose sign of curvature is set by the sign of

$$\Lambda_{\text{eff}} \equiv \frac{\Lambda + 4\pi G_N m^2 \bar{\phi}^2}{1 - 8\pi G_N \xi \bar{\phi}^2}. \tag{119}$$

We note that even for a theory with zero initial cosmological constant, $\Lambda = 0$, the constrained saddle has positive curvature induced by the massive scalar. We also note that as $\bar{\phi}^2$ approaches $(8\pi G_N \xi)^{-1}$ from below, the effect is to "shrink" the classical spacetime. We will see the effect of this on the semi-classical path-integral below.

The on-shell action of this constrained solution

$$I_\xi[\bar{\phi}, \bar{g}] = -\text{vol}[\bar{g}]\left(\frac{1}{n-2}\right)\left(\frac{\Lambda}{4\pi G_N} + m^2\bar{\phi}^2\right), \tag{120}$$

where $\text{vol}[\bar{g}]$ is the Euclidean spacetime volume determined by $\bar{g}$. The nature of this volume depends on the sign of $\Lambda_{\text{eff}}$. Let us focus on $\Lambda_{\text{eff}} > 0$ and investigate what happens as $\bar{\phi}^2$ approaches $(8\pi G_N \xi)^{-1}$ from below.

In this case $\bar{g}_{\mu\nu}$ is the metric of a round $n$-sphere of radius $\ell_{\text{eff}}$ and has a volume

$$\text{vol}[\bar{g}] = V_n \ell^n_{\text{eff}}, \tag{121}$$

where we recall $V_n = \frac{2\pi^{\frac{n+1}{2}}}{\Gamma\left(\frac{n+1}{2}\right)}$ is the volume of a unit $n$-sphere. After some massaging, the on-shell action is

$$I_\xi[\bar{\phi}, \bar{g}] = -\frac{V_{n-2}}{4G_N}\left(\frac{(n-1)(n-2)}{2}\right)^{\frac{n-2}{2}}\left(1 - 8\pi G_N \xi \bar{\phi}^2\right)^{n/2}\left(\Lambda + 4\pi G_N m^2 \bar{\phi}^2\right)^{\frac{2-n}{2}}. \tag{122}$$

We see that as $\bar{\phi}^2$ approaches $(8\pi G_N \xi)^{-1}$ the action approaches zero.

---

[14]The saddle-point value of $\lambda$ depends implicitly on $\bar{\phi}$ through $\bar{R}$. We will choose the $\lambda$ contour to run through $\bar{\lambda} + i\mathbb{R}$ and the independent integration over the imaginary axis correctly implements the constraint.

Returning now to our original path-integral and approximating $Z_{\text{con}}$ by its on-shell action

$$Z_{\text{matter+grav}} \approx \int d(\bar{\phi}^2)\, e^{\frac{V_{n-2}}{4G_N}(1-8\pi G_N \xi \bar{\phi}^2)^{n/2}(\Lambda + 4\pi G_N m^2 \bar{\phi}^2)^{(2-n)/2}} \times z_{\text{1-loop}}[\bar{\phi}^2], \qquad (123)$$

where $z_{\text{1-loop}}$ schematically contains the contributions from the Gaussian integrations about the constrained saddle. As $\bar{\phi}^2 \to (8\pi G_N \xi)^{-1}$, the contribution from the on-shell action becomes unity and the path-integral is dominated by the one-loop contribution, $z_{\text{1-loop}}[\bar{\phi}^2]$. We regard this as a manifestation of the breakdown of the semi-classical path-integral in the gravitational sector. Additionally, from inspection of Eq. (110), $G_N/(1-8\pi G_N \xi \phi^2)$ plays the role of an effective coupling constant for the gravitational sector: This is also the regime where fluctuations about the constrained saddle-point, appearing in $z_{\text{1-loop}}$, become strongly coupled.

In the regime beyond the critical value, $(8\pi G_N \xi)^{-1}$, lay dragons. The naïve solution of the saddle-point equations, Eq. (115), is a space with constant negative curvature as a result of the negative effective cosmological constant, $\Lambda_{\text{eff}} < 0$. While this does not necessarily imply an infinite Euclidean volume[15] (and so the constant mode of $\phi^2$ may remain a sensible modulus), the Cartan-Hadamard theorem implies that if such spaces are simply-connected they must be non-compact (so if they have finite volume they must be cusped). At a conservative level, without incorporating topology change effects, such solutions will not satisfy the boundary conditions of the original gravity path-integral and so cannot be kept as saddles in $Z_{\text{con}}$. True saddles may still exist for $Z_{\text{con}}$ when $\bar{\phi}^2 > (8\pi G_N \xi)^{-1}$ but must incorporate new topology[16] compared to the $\bar{\phi}^2 < (8\pi G_N \xi)^{-1}$ saddles.

Lastly, we will comment on the situation with negative coupling, $\xi < 0$. The analysis of the constrained saddles still holds for this sign of coupling, giving a homogeneous space of curvature set by Eq. (118) and an effective cosmological constant set by Eq. (119). For a positive bare $\Lambda > 0$ the constrained saddle remains a compact space with an action set by Eq. (122) for all values of $\bar{\phi}^2$. It is at the level of the path-integral that the analysis deviates. As $\bar{\phi}^2$ grows, so does $\ell_{\text{eff}}^2$ and the gravitational action moves to weaker coupling. This suggests that the semi-classical analysis is better behaved for a large field values. However a cursory analysis of Eq. (123) also displays the following pathologies: interpreting $e^{-I_\xi[\bar{\phi},\bar{g}]}$ as the leading contribution to an unnormalised probability density for finding a field value $\bar{\phi}^2$, this theory is driven towards larger field values. A larger problem however is that this probability density is not only unnormalised, it is non-normalisable: The integral over $d(\bar{\phi}^2)$ in Eq. (123) diverges! The most natural resolution of this divergence is to impose a cutoff on the upper end of this integral i.e. again to regard the theory as an effective field theory for states with a cutoff on $|\phi^2|$. As opposed to the case with positive $\xi$, there is no natural reason to associate this field cutoff with the Planck scale, however.

## 6 Conclusion

In this work we explored the effects of non-minimal coupling to null energy, in both classical and quantum physics. The main motivation is to answer the question of whether or not the non-minimal coupling with gravity is sufficient to allow exotic physics.

Superficially the answer is yes; non-minimal coupling allows violations of the null energy condition (NEC) even classically, while in quantum field theory (QFT) the construction of states with infinite negative null energy is possible. Meanwhile, performing a conformal transformation and a field redefinition we can go from the action of the non-minimally coupled scalar (Jordan frame) to the one with minimal coupling (Einstein frame) where none of these problems are present.

---

[15]Even the infinite volume solutions could be assigned a renormalised volume, e.g. [40].

[16]Namely a non-trivial fundamental group.

Here we adopt the view that the physically relevant question is if the two frames lead to similar spacetime geometries. In the effective field theory (EFT) framework we are using, the metrics are necessarily close. Moreover, traversable wormholes are only possible if the effective average null energy condition (ANEC) is violated. Classically the violation of the effective ANEC is restricted to large field values, while semiclassically, the smeared null energy admits a lower bound dependent on the cutoff. Bounded field values additionally lead to a semiclassical proof of ANEC.[17] Additionally, the transformation between the two frames leaves the Einstein frame theory as an interacting QFT, for which there are no known state-independent bounds in $n > 2$. Finally, path-integral considerations support the EFT treatment as large field values lead to a breakdown of the effective theory, either through a tower of irrelevant interactions in the Einstein frame, or a breakdown of the semi-classical path integral in the gravitational sector.

A natural extension of this work is the study of energy conditions in self-interacting field theories. That would allow for a more complete treatment of the lower bounds of the smeared null energy in the Einstein frame. Quantum energy inequalities (QEIs) for self-interacting fields are notoriously difficult to treat, with positive results only in two-dimensional models [30,41,42], and in the case of the ANEC [43,44] in Minkowski space. However, a perturbative study of the $\lambda \phi^4$ model, the leading term that appears in the Einstein frame, seems promising.

Lastly we have focused on a particular type of state-dependent QEI where the state-dependence takes the form of an expectation value. More general state-dependent inequalities may use more non-linear, entropic, features of a state. The quintessential example of this is the quantum null energy condition (QNEC) which has been generally proven in a Minkowski background [45,46]. Given that the QNEC follows from the quantum focusing conjecture [47], via Raychaudhuri's equation, it is interesting to speculate whether QNEC must be modified in the presence of non-minimal coupling (NMC) in the Jordan frame. Even in a Minkowski background, where the residue of NMC can be viewed as an improvement term to the free stress tensor, the NMC theory may evade the known proofs of QNEC which either only apply to interacting theories [46], or rely on the form of the stress tensor as a boost operator [45]. Given the ubiquity of NMC terms in Jordan frame effective actions, we view the status of the QNEC in the presence of NMC as an interesting and potentially important open question.

# Acknowledgments

The authors would like to thank Diego Hofman, John Stout, Ken Olum, and Manus Visser for helpful conversations. JRF thanks the University of Amsterdam for accommodation where some of this work was completed.

**Funding information** JRF has been partially supported by STFC consolidated grant ST/T000694/1, the ERC starting grant GenGeoHoloIC, and by Simons Foundation Award number 620869. BF and E-AK were supported by the ERC Consolidator Grant QUANTIVIOL. BF was additionally supported by Heising-Simons Foundation "Observational Signatures of Quantum Gravity" collaboration grant 2021-2817. This work is part of the $\Delta$-ITP consortium, a program of the NWO that is funded by the Dutch Ministry of Education, Culture and Science (OCW). DPS is supported by the EPSRC studentship grant EP/W524475/1.

---

[17]We presented a proof for Minkowski spacetime but we expect it could be extended to spacetimes with finite curvature as in Ref. [25].

# A    Field redefinition for general coupling

In this appendix we comment on a more general form of the non-minimal coupling and show that it can be reduced to an Einstein frame with a suitable field redefinition. We will start with an action

$$S = \int d^n x \sqrt{-g} \left[ \frac{(R - 2\Lambda)}{16\pi G_N} - \frac{1}{2}(\nabla\phi)^2 - \mathcal{A}(\phi)R - V(\phi) \right], \qquad (A.1)$$

where $\mathcal{A}$ is a function of $\phi$, which we might regard as the first term of an EFT organized by the usual derivative expansion. We now perform a simultaneous conformal transformation and field redefinition

$$\tilde{g}_{\mu\nu} = \Omega^2(\phi(x))g_{\mu\nu}, \qquad \tilde{\phi}(x) = F(\phi(x)), \qquad (A.2)$$

with the aim of (i) making the coupling in front of the new Ricci scalar, $\tilde{R}$, appear solely as $1/(16\pi G_N)$, and (ii) making the kinetic term of $\tilde{\phi}$ canonical. Focussing first on the transformation of $R$ into $\tilde{R}$, we have

$$S = \int d^n x \sqrt{-\tilde{g}}\,\Omega^{2-n}\frac{1 - 16\pi G_N \mathcal{A}}{16\pi G_N}\left(\tilde{R} + 2(n-1)\tilde{\nabla}^2\log\Omega - (n-2)(n-1)(\tilde{\nabla}\log\Omega)^2\right) + \dots, \qquad (A.3)$$

and so canonicalizing the Ricci term requires

$$\Omega(\phi) = (1 - 16\pi G_N \mathcal{A}(\phi))^{\frac{1}{n-2}}. \qquad (A.4)$$

Upon imposing (A.4), the second term of (A.3) is a total derivative which we will drop, while the third term will combine with kinetic term of $\tilde{\phi}$ to give

$$S = \int d^n x \sqrt{-\tilde{g}}\left(\frac{\tilde{R}}{16\pi G_N} - \frac{1}{2}\frac{\left(1 - 16\pi G_N(\mathcal{A} - \frac{2}{\xi_c}(\mathcal{A}')^2)\right)}{(1 - 16\pi G_N \mathcal{A})^2}\tilde{g}^{\mu\nu}\partial_\mu\phi\,\partial_\nu\phi + \dots\right), \qquad (A.5)$$

where $\xi_c = \frac{n-2}{4(n-1)}$ is the conformal coupling. Noting that $\partial_\mu\phi = \left(F^{-1}\right)'\partial_\mu\tilde{\phi}$, we can canonicalize the kinetic term for $\tilde{\phi}$ by solving the differential equation

$$F'(x) = (1 - 16\pi G_N \mathcal{A}(x))^{-1}\sqrt{1 - 16\pi G_N\left(\mathcal{A} - \frac{2}{\xi_c}(\mathcal{A}'(x))^2\right)}. \qquad (A.6)$$

The result is a minimally coupled action with a canonical scalar kinetic term

$$S = \int d^n x \sqrt{-\tilde{g}}\left[\frac{\tilde{R}}{16\pi G_N} - \frac{1}{2}(\tilde{\nabla}\tilde{\phi})^2 - \tilde{V}(\tilde{\phi})\right], \qquad (A.7)$$

and with an effective potential

$$\tilde{V}(\tilde{\phi}) = \Omega(F^{-1}(\tilde{\phi}))^{-n}\left(\frac{\Lambda}{8\pi G_N} + V(F^{-1}(\tilde{\phi}))\right), \qquad (A.8)$$

where $\Omega$ and $F$ satisfy (A.4) and (A.6), respectively. Notice that all of the details of the coupling, $\mathcal{A}(\phi)$, have been moved to the potential, $\tilde{V}$. Thus, at least classically, the NEC is obeyed in this general Einstein frame.

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
