# Peer review of "Non-minimal coupling, negative null energy, and effective field theory"

_SciPost Physics, doi:SciPost Phys. 16, 119 (2024)_

## Round 1 · Referee Report · Anonymous (Referee 1) · 2024-2-29

Report

I recommend for publication.
I have few queries which I have written in my report. It would be nice if they could be addressed.

Attachment

  • validity: -
  • significance: -
  • originality: -
  • clarity: -
  • formatting: -
  • grammar: -

Author:  Diego Pardo Santos  on 2024-03-30  [id 4384]

(in reply to Report 1 on 2024-02-29)

Dear Referee,

Thank you for your suggestions for improvement. We would like to take the chance to address some specific confusions and opportunities for improving the clarity of the article as you pointed out.

1) In reply to the comment: “I feel it would be nice to have some discussion on how the analysis could be extended to any general non-minimal coupling.”. We have focussed here on the simplest form of non-minimal coupling because this is first, and most relevant term in our EFT framework (which is organized by both a derivative and field value expansion). Thus it is already important to investigate the extent of energy condition violations for this form of coupling. Regardless, we are grateful for making this point; in response we have expanded the introduction (namely the third paragraph) to emphasize the importance of this point and our results.  Additionally we have added an appendix discussing more general curvature couplings and show that they have a suitable Einstein frame where the NEC is classically satisfied.  We have added several comments to the paper referring to this, where appropriate (see footnote 4 on page 5, the paragraph under equation (23), and the paragraph under equation (102)). Lastly, one of the main results of this manuscript, the lower bounding of the integrated null energy density by the Wick square of the scalar field is cast as a quantum energy inequality, equation (73), that holds on curved backgrounds, assuming the existence of a suitable reference state. We have added a small paragraph (second paragraph under equation (73)) emphasizing the generality of this result.

2) In reply to the comment: “The definition of a field is a matter of choice and the final physical outcome should not depend on how we choose to define fields. [...] However, it seems that the invariance of physics under different choices of field variables is a very general principle, independent of whether we want to describe the physics classically or quantum mechanically, as an EFT valid only for small values of fields or as an exact theory valid everywhere. From this perspective I am confused if the null energy condition is classically not violated in Einstein frame, then why it is violated in Jordan frame at large values of the field.”. The operators in different frames are distinct operators which is why the stress tensor of the Einstein frame can (classically) satisfy the NEC while that of the Jordan does not. As we state in the paragraph surrounding equation (4) which operator one considers physically relevant is determined by the spacetime that they naturally probe. Regardless, at least classically, there remains a clear map between probes in the two frames, which follows from the field redefinition. We agree with  that the physics of the two frames are exactly equivalent quantum theories (assuming they exist as exact theories), which we formalize in Section 5 as a change of path-integration variables. However the map relating operators in one frame is complicated by Jacobians from the path-integral measure.  We argue in that section that in the EFT framework these corrections remain controlled. We feel that many of the above comments have been already addressed within the body of the manuscript, but to improve clarity, we have added sentences in the beginning of the paragraph before equation (3), the end of the second paragraph after equation (4), and in the second paragraph of section 5, emphasizing that the stress tensors of the frames are distinct operators with a well-defined map within the EFT framework.

List of changes: 1) Expanded paragraph under equation (2) to emphasize the relevance of the non-minimal coupling vs. generic curvature couplings. 2) Expanded first sentence of paragraph above equation (3) to emphasize that the stress-tensors of the two frames are distinct operators. 3) Added two sentences to the beginning of the paragraph above equation (4) to highlight the importance of the problem and our results. 4) Added sentence to end of second paragraph on page 4 to emphasize that the relation between frames remains controlled in the EFT framework. 5) Added footnote 4 on page 5 to refer to Appendix A regarding generic curvature couplings. 6) Added sentence to paragraph under equation (23) regarding generic curvature couplings and referring to Appendix A. 7) Added a paragraph under Eq.(73) to discuss curvature. 8) Expanded the second paragraph of section 5 to emphasize the difference between the two frames and emphasize the role of the EFT in controlling quantum corrections to the map between frames. 9) Added sentence below equation (102) regarding generic curvature couplings and referring to Appendix A. 10) Fixed a typo in equation (119). 11) Added an appendix (Appendix A) illustrating the existence of Einstein frame with a classically satisfied NEC for generic curvature couplings and constructing the field redefinition that relates the operators between the frames.

---

## Round 1 · Referee Report · Anonymous (Referee 2) · 2024-3-2

Report

Dear Editor,

In this paper, the authors investigated the null energy condition (both classically and for quantum field theory) in the presence of non-minimal coupling. They find that classically it can be violated in the Jordan frame while it is satisfied in the Einstein frame. This is not very surprising, as under frame transformation it is not clear that the observables will also transform in the same way. Is it true for the stress tensor? i.e. under the frame transformation equation (16) reduces to equation (24) ?. If not then, this apparent violation of NEC in one frame is not very surprising.

The authors investigated what happens in quantum field theory. They found that in the Jordan frame, there can be states with infinite null energy. But in the Einstein frame, there are no such subtleties. Then gave some EFT arguments to explain this issue. They make some very generic claims based on their particular model and that too for a Minkowski spacetime. A bit more general examples are needed to make sense of their claim. At this point, it is of limited novelty. In light of this, I will recommend this to be published in SciPost PhysicsPhysics core given its narrow scope instead of SciPost Physics.

Best,
Referee
  • validity: ok
  • significance: ok
  • originality: ok
  • clarity: ok
  • formatting: reasonable
  • grammar: -

Author:  Diego Pardo Santos  on 2024-03-30  [id 4385]

(in reply to Report 2 on 2024-03-02)

Dear Referee,

We thank you for your suggestions for improvement. We would like to take the chance to address some specific confusions and opportunities for improving the clarity of the article as you pointed out.

1) In reply to the comment:  “They make some very generic claims based on their particular model and that too for a Minkowski spacetime. A bit more general examples are needed to make sense of their claim.” We have focussed here on the simplest form of non-minimal coupling because this is first, and most relevant term in our EFT framework (which is organized by both a derivative and field value expansion). Thus it is already important to investigate the extent of energy condition violations for this form of coupling. Regardless, we are grateful for making this point; in response we have expanded the introduction (namely the third paragraph) to emphasize the importance of this point and our results.  Additionally we have added an appendix discussing more general curvature couplings and show that they have a suitable Einstein frame where the NEC is classically satisfied.  We have added several comments to the paper referring to this, where appropriate (see footnote 4 on page 5, the paragraph under equation (23), and the paragraph under equation (102)). Lastly, one of the main results of this manuscript, the lower bounding of the integrated null energy density by the Wick square of the scalar field is cast as a quantum energy inequality, equation (73), that holds on curved backgrounds, assuming the existence of a suitable reference state. We have added a small paragraph (second paragraph under equation (73)) emphasizing the generality of this result.

2) In reply to the comment: “They find that classically it can be violated in the Jordan frame while it is satisfied in the Einstein frame. This is not very surprising, as under frame transformation it is not clear that the observables will also transform in the same way.”  It is indeed true that the operators in different frames are distinct operators which is why the stress tensor of the Einstein frame can (classically) satisfy the NEC while that of the Jordan does not. As we state in the paragraph surrounding equation (4) which operator one considers physically relevant is determined by the spacetime that they naturally probe. Regardless, at least classically, there remains a clear map between probes in the two frames, which follows from the field redefinition. The physics of the two frames are exactly equivalent quantum theories (assuming they exist as exact theories), which we formalize in Section 5 as a change of path-integration variables. However the map relating operators in one frame is complicated by Jacobians from the path-integral measure.  We argue in that section that in the EFT framework these corrections remain controlled. We feel that many of the above comments have been already addressed within the body of the manuscript, but to improve clarity, we have added sentences in the beginning of the paragraph before equation (3), the end of the second paragraph after equation (4), and in the second paragraph of section 5, emphasizing that the stress tensors of the frames are distinct operators with a well-defined map within the EFT framework.

3) Lastly to address the issue of novelty. The specific non-minimal coupling we consider in this paper is very natural to consider as the first relevant term in a series of curvature couplings. As such it appears in many EFT contexts (e.g. citations [1]-[4] of the manuscript); and importantly contains the case of the conformally coupled scalar. We have commented on the applicability to general couplings above in 1). To this end the study of energy conditions for this coupling has strong merit and we feel that our results give a novel new perspective on the regime of validity of this theory as an EFT. Sections 3 and 4, while admittedly focussed on Minkowski space, illustrate two broadly important points: (i) that non-minimal coupling can badly violate smeared energy conditions even in the simplest of scenarios (in Section 3) and (ii) to prove a relation between DSNEC violation and large field values (in Section 4). As mentioned in point 1) of our rebuttal above, this relation follows from a quantum energy inequality that holds in a general class of curved backgrounds. We have added a sentence at the beginning of the paragraph before equation (4) to emphasize the importance of our results.

In light of the above points and the changes made to the manuscript, we maintain that the manuscript be published in SciPost Physics.

List of changes: 1) Expanded paragraph under equation (2) to emphasize the relevance of the non-minimal coupling vs. generic curvature couplings. 2) Expanded first sentence of paragraph above equation (3) to emphasize that the stress-tensors of the two frames are distinct operators. 3) Added two sentences to the beginning of the paragraph above equation (4) to highlight the importance of the problem and our results. 4) Added sentence to end of second paragraph on page 4 to emphasize that the relation between frames remains controlled in the EFT framework. 5) Added footnote 4 on page 5 to refer to Appendix A regarding generic curvature couplings. 6) Added sentence to paragraph under equation (23) regarding generic curvature couplings and referring to Appendix A. 7) Added a paragraph under Eq.(73) to discuss curvature. 8) Expanded the second paragraph of section 5 to emphasize the difference between the two frames and emphasize the role of the EFT in controlling quantum corrections to the map between frames. 9) Added sentence below equation (102) regarding generic curvature couplings and referring to Appendix A. 10) Fixed a typo in equation (119). 11) Added an appendix (Appendix A) illustrating the existence of Einstein frame with a classically satisfied NEC for generic curvature couplings and constructing the field redefinition that relates the operators between the frames.

---

## Round 2 · Referee Report · Anonymous (Referee 2) · 2024-3-31

Report

I am happy with the changes made by the authors and recommending the revised version for the publication.

---

## Round 2 · Referee Report · Anonymous (Referee 1) · 2024-4-3

Report

I am happy with the changes and new explanations introduced by the authors. I have no further queries. I recommend it for publication.

---

## Round 2 · Author Response

Dear Editor-in-charge.

We thank the referees for their suggestions for improvement. We would like to take the chance to address some specific confusions and opportunities for improving the clarity of the article as pointed out by the referees.

1) In reply to the comment from referee 1, “I feel it would be nice to have some discussion on how the analysis could be extended to any general non-minimal coupling.” and similarly referee 2, “They make some very generic claims based on their particular model and that too for a Minkowski spacetime. A bit more general examples are needed to make sense of their claim.” We have focussed here on the simplest form of non-minimal coupling because this is first, and most relevant term in our EFT framework (which is organized by both a derivative and field value expansion). Thus it is already important to investigate the extent of energy condition violations for this form of coupling. Regardless, we are grateful to the referees for making this point; in response we have expanded the introduction (namely the third paragraph) to emphasize the importance of this point and our results.  Additionally we have added an appendix discussing more general curvature couplings and show that they have a suitable Einstein frame where the NEC is classically satisfied.  We have added several comments to the paper referring to this, where appropriate (see footnote 4 on page 5, the paragraph under equation (23), and the paragraph under equation (102)). Lastly, one of the main results of this manuscript, the lower bounding of the integrated null energy density by the Wick square of the scalar field is cast as a quantum energy inequality, equation (73), that holds on curved backgrounds, assuming the existence of a suitable reference state. We have added a small paragraph (second paragraph under equation (73)) emphasizing the generality of this result.
2) In reply to the comment from referee 1, “The definition of a field is a matter of choice and the final physical outcome should not depend on how we choose to define fields. [...] However, it seems that the invariance of physics under different choices of field variables is a very general principle, independent of whether we want to describe the physics classically or quantum mechanically, as an EFT valid only for small values of fields or as an exact theory valid everywhere. From this perspective I am confused if the null energy condition is classically not violated in Einstein frame, then why it is violated in Jordan frame at large values of the field.” and similarly from referee 2, “They find that classically it can be violated in the Jordan frame while it is satisfied in the Einstein frame. This is not very surprising, as under frame transformation it is not clear that the observables will also transform in the same way.”  It is indeed true that the operators in different frames are distinct operators which is why the stress tensor of the Einstein frame can (classically) satisfy the NEC while that of the Jordan does not. As we state in the paragraph surrounding equation (4) which operator one considers physically relevant is determined by the spacetime that they naturally probe. Regardless, at least classically, there remains a clear map between probes in the two frames, which follows from the field redefinition. We agree with referee 1 that the physics of the two frames are exactly equivalent quantum theories (assuming they exist as exact theories), which we formalize in Section 5 as a change of path-integration variables. However the map relating operators in one frame is complicated by Jacobians from the path-integral measure.  We argue in that section that in the EFT framework these corrections remain controlled. We feel that many of the above comments have been already addressed within the body of the manuscript, but to improve clarity, we have added sentences in the beginning of the paragraph before equation (3), the end of the second paragraph after equation (4), and in the second paragraph of section 5, emphasizing that the stress tensors of the frames are distinct operators with a well-defined map within the EFT framework.
3) Lastly to address the issue of novelty raised by referee 2. The specific non-minimal coupling we consider in this paper is very natural to consider as the first relevant term in a series of curvature couplings. As such it appears in many EFT contexts (e.g. citations [1]-[4] of the manuscript); and importantly contains the case of the conformally coupled scalar. We have commented on the applicability to general couplings above in 1). To this end the study of energy conditions for this coupling has strong merit and we feel that our results give a novel new perspective on the regime of validity of this theory as an EFT. Sections 3 and 4, while admittedly focussed on Minkowski space, illustrate two broadly important points: (i) that non-minimal coupling can badly violate smeared energy conditions even in the simplest of scenarios (in Section 3) and (ii) to prove a relation between DSNEC violation and large field values (in Section 4). As mentioned in point 1) of our rebuttal above, this relation follows from a quantum energy inequality that holds in a general class of curved backgrounds. We have added a sentence at the beginning of the paragraph before equation (4) to emphasize the importance of our results.

In light of the above points and the changes made to the manuscript, we maintain that the manuscript be published in SciPost Physics.

---

## Round 2 · List of Changes

List of changes: 1) Expanded paragraph under equation (2) to emphasize the relevance of the non-minimal coupling vs. generic curvature couplings. 2) Expanded first sentence of paragraph above equation (3) to emphasize that the stress-tensors of the two frames are distinct operators. 3) Added two sentences to the beginning of the paragraph above equation (4) to highlight the importance of the problem and our results. 4) Added sentence to end of second paragraph on page 4 to emphasize that the relation between frames remains controlled in the EFT framework. 5) Added footnote 4 on page 5 to refer to Appendix A regarding generic curvature couplings. 6) Added sentence to paragraph under equation (23) regarding generic curvature couplings and referring to Appendix A. 7) Added a paragraph under Eq.(73) to discuss curvature. 8) Expanded the second paragraph of section 5 to emphasize the difference between the two frames and emphasize the role of the EFT in controlling quantum corrections to the map between frames. 9) Added sentence below equation (102) regarding generic curvature couplings and referring to Appendix A. 10) Fixed a typo in equation (119). 11) Added an appendix (Appendix A) illustrating the existence of Einstein frame with a classically satisfied NEC for generic curvature couplings and constructing the field redefinition that relates the operators between the frames.

---

## Editorial Decision

published